# The preRC protein ORCA organizes heterochromatin by assembling histone H3 lysine 9 methyltransferases on chromatin

Sumanprava Giri[1], Vasudha Aggarwal[2], Julien Pontis[3], Zhen Shen[1], Arindam Chakraborty[1], Abid Khan[1], Craig Mizzen[1], Kannanganattu V Prasanth[1], Slimane Ait-Si-Ali[3], Taekjip Ha[2,4], Supriya G Prasanth[1]*

[1]Department of Cell and Developmental Biology, University of Illinois at Urbana-Champaign, Champaign, United States; [2]Center for Biophysics and Computational Biology, University of Illinois at Urbana-Champaign, Champaign, United States; [3]Université Paris Diderot, Sorbonne Paris Cité, Laboratoire Epigénétique et Destin Cellulaire, UMR7216, Centre National de la Recherche Scientifique, Paris, France; [4]Department of Physics, Howard Hughes Medical Institute, University of Illinois at Urbana-Champaign, Champaign, United States

**Abstract** Heterochromatic domains are enriched with repressive histone marks, including histone H3 lysine 9 methylation, written by lysine methyltransferases (KMTs). The pre-replication complex protein, origin recognition complex-associated (ORCA/LRWD1), preferentially localizes to heterochromatic regions in post-replicated cells. Its role in heterochromatin organization remained elusive. ORCA recognizes methylated H3K9 marks and interacts with repressive KMTs, including G9a/GLP and Suv39H1 in a chromatin context-dependent manner. Single-molecule pull-down assays demonstrate that ORCA-ORC (Origin Recognition Complex) and multiple H3K9 KMTs exist in a single complex and that ORCA stabilizes H3K9 KMT complex. Cells lacking ORCA show alterations in chromatin architecture, with significantly reduced H3K9 di- and tri-methylation at specific chromatin sites. Changes in heterochromatin structure due to loss of ORCA affect replication timing, preferentially at the late-replicating regions. We demonstrate that ORCA acts as a scaffold for the establishment of H3K9 KMT complex and its association and activity at specific chromatin sites is crucial for the organization of heterochromatin structure.

*For correspondence: supriyap@ life.illinois.edu

## Introduction

Origin recognition complex-associated (ORCA/LRWD1), a protein required for the initiation of DNA replication, preferentially localizes to heterochromatic regions in post-replicated cells (*Bartke et al., 2010*; *Shen et al., 2010*; *Vermeulen et al., 2010*). We and others have demonstrated that ORCA and ORC associate with centromeric and telomeric heterochromatin in mammalian cells (*Shen et al., 2010*). Furthermore, using a stable isotope labeling by amino acids in cell culture (SILAC)-based proteomic approach, ORCA-ORC complex has been shown to bind the repressive histone lysine methylation marks, specifically H3K9me3, H3K27me3, and H4K20me3 (*Bartke et al., 2010*; *Vermeulen et al., 2010*) that are known to be enriched at heterochromatic sites. ORCA contains a WD domain, a structure known to interact with nucleosomes/histones (*Wysocka et al., 2005*). We have previously demonstrated that the WD domain of ORCA is crucial for its binding to heterochromatin. Furthermore, ORCA is critical for stabilizing ORC binding to chromatin

**eLife digest** The genetic material inside cells is contained within molecules of DNA. In animals and other eukaryotes, the DNA is tightly wrapped around proteins called histones to form a compact structure known as chromatin. There are two forms of chromatin: loosely packed chromatin tends to contain genes that are highly active in cells, while tightly packed chromatin—called heterochromatin—tends to contain less-active genes.

How tightly DNA is packed in chromatin can be changed by adding small molecules known as methyl tags to individual histone proteins. Enzymes called KMTs are responsible for attaching these methyl tags to a specific site on the histones. Before a cell divides, it duplicates its DNA and these methyl tags, so that they can be passed onto the newly formed cells. This enables the new cells to 'remember' which genes were inactive or active in the original cell. A protein known as ORCA associates with heterochromatin, but it is not clear what role it plays in controlling the structure of chromatin.

Giri et al. studied ORCA and the KMTs in human cells. The experiments show that ORCA recognizes the methyl tags and binds to the KMTs in regions of heterochromatin, but not in regions where the DNA is more loosely packed. Next, Giri et al. used a technique called single-molecule pull-down, which is able to identify individual proteins within a group. These experiments showed that several KMT enzymes can bind to a single ORCA protein at the same time. ORCA stabilizes the binding of KMTs to chromatin, which enables the KMTs to modify the histones within it.

Cells lacking ORCA had fewer methyl tags on their histones, which altered the structure of the chromatin. This also affected the timing with which DNA copying takes place in cells before the cell divides. Giri et al.'s findings suggest that ORCA acts as a scaffold for the KMTs and that its activity at specific sites on chromatin is important for the organization of heterochromatin. The next step is to identify the exact regions in the genome where the timing of DNA copying is regulated by ORCA.

(*Shen et al., 2010*). ORC, a hetero-hexameric complex, in addition to serving as the landing pad for the assembly of pre-replicative complex at the origins of DNA replication, participates in sister chromatid cohesion, heterochromatin organization, and chromosome segregation (*Bell et al., 1993*; *Shimada et al., 2002*; *Sasaki and Gilbert, 2007*). In metazoans, ORC also facilitates the association of heterochromatin protein 1 (HP1) to the H3K9me3-containing pericentric heterochromatin (*Pak et al., 1997*; *Prasanth et al., 2004*, *2010*). Thus, it is obvious that ORC-ORCA complex associates with heterochromatin, but the mechanism underlying the recruitment of this multiprotein complex to the condensed chromatin and the functional relevance of such association has remained elusive for decades.

Histone lysine methylation, catalyzed by lysine methyltransferases (KMTs), plays key roles in the epigenetic regulation of chromatin organization, transcription, and replication (*Black et al., 2012*). Methylation of H3K9 is an abundant and stable modification and is an important regulator of heterochromatin formation, gene silencing, and DNA methylation (*Martin and Zhang, 2005*). The methyl modifications on H3K9 exist in distinct mono-, di-, and tri-methyl states (H3K9me1, H3K9me2, and H3K9me3, respectively), with each responsible for governing distinct cellular functions. In general, H3K9me1 and H3K9me2 are associated with gene expression/repression at euchromatic regions, whereas the H3K9me3, enriched at pericentric heterochromatin, is required for heterochromatin assembly and gene silencing (*Martin and Zhang, 2005*). The major KMTs catalyzing these modifications are G9a and GLP, responsible for H3K9me2 (*Tachibana et al., 2001*; *Rice et al., 2003*; *Shinkai and Tachibana, 2011*); SETDB1, which establishes H3K9 di- and tri-methylation in euchromatin (*Schultz et al., 2002*) and Suv39H1/H2 that establishes H3K9me3 from mono- or di-methylated H3K9 (*Rea et al., 2000*; *Peters et al., 2003*). While the idea of G9a and Suv39H1 acting in distinct, primarily in non-overlapping chromatin contexts held sway for a long time, this concept has been recently challenged by the discovery of a complex consisting of multiple H3K9 KMTs (*Fritsch et al., 2010*). The multimeric complex contains all four H3K9 KMTs G9a, GLP, Suv39H1, and SETDB1 and is recruited to both pericentromeric heterochromatin and promoter of a set of G9a-repressed genes where it aids in gene repression by maintaining H3K9me2 and H3K9me3 marks (*Fritsch et al., 2010*). Furthermore, destabilizing even one of these KMTs resulted in the disintegration of the

multimeric complex and loss of the enzymatic activity of this complex (*Fritsch et al., 2010*). How this multimeric KMTs complex is recruited to specific chromatin sites remained to be determined. In the broader context, the functional significance of the crosstalk between chromatin modifying and replication machineries has remained largely unexplored.

Here, we demonstrate that ORCA associates with H3K9 KMTs in a chromatin context-dependent manner. By using a highly sensitive and quantitative single-molecule pull-down (SiMPull) approach (*Jain et al., 2011*; *Shen et al., 2012*), we demonstrate that ORCA preferentially binds to H3K9me3 and ORCA-ORC, and multiple H3K9 KMTs exist in a single complex. Furthermore, ORCA is required for the formation and/or maintenance of the H3K9 KMT complex. Our results indicate that ORCA is required for the integrity of global chromatin architecture. In the absence of ORCA, human cells show alterations in the binding and activity of KMTs at sites enriched for these factors with concomitant reduction in H3K9me2 and H3K9me3 marks. Finally, we observe that the cells lacking ORCA display abnormal heterochromatin organization and alteration in the replication timing, specifically at the late-replicating regions. We propose that ORCA is a scaffold protein that is required for the establishment as well as maintenance of heterochromatin.

## Results

### ORCA interacts with H3K9 KMTs

In order to address if ORCA interacts with the machinery that causes the establishment of heterochromatin, we used a candidate approach to investigate the interaction of ORCA with individual H3K9 KMTs that catalyze H3K9 repressive modifications. We observed robust interaction of endogenous ORCA with endogenous G9a and Suv39H1 (*Figure 1Aa,Ab*, *Figure 1—figure supplement 1Aa,Ab*). 1.31% of total G9a was found to be in a complex with ORCA. Quantitation was based on the amount of G9a immunoprecipitated with ORCA (based on 100% efficiency of ORCA IP, *Figure 1—figure supplement 1B*) (n = 7). Similarly, 1.44% of total Suv39H1 was in a complex with ORCA (n = 4). Note that only about 0.2% of the endogenous H3K9 KMTs co-purified with Suv39H1 (*Fritsch et al., 2010*). Co-immunoprecipitation (co-IP) using T7-tagged ORCA and Flag-tagged H3K9 KMTs revealed interaction of ORCA with G9a, GLP, and Suv39H1, all enzymes involved in the establishment of heterochromatin (*Figure 1Ba,Bb*). In addition, we carried out IP from cell lines stably expressing Flag-tagged-G9a or GLP. IP from nuclear extracts using Flag antibody to determine the association of endogenous ORCA with the KMTs. ORCA along with Orc2 and MCMs was found to interact with the KMTs (*Figure 1—figure supplement 1C*). However, ORCA did not associate with the arginine methyltransferase PRMT5 (*Figure 1—figure supplement 1D*), showing the specificity of the interactions.

In order to show functional co-recruitment of ORCA and the H3K9 KMTs, we used an in vivo cell system (CLTon) that uses a 200 copy transgene array-containing lac operator repeats stably integrated into human osteosarcoma (U2OS) cells as a single heterochromatic locus that can be visualized by Cherry-lac repressor (LacI). Furthermore, transcriptional activation using doxycycline (DOX) causes the decondensation of the locus (*Janicki et al., 2004*; *Shen et al., 2010*). We tethered the triple fusion proteins of YFP-LacI-KMTs to the heterochromatic locus and examined if these enzymes could recruit ORCA to the locus. This approach corroborated the interaction of ORCA with G9a (*Figure 1C*).

We next examined whether ORCA and H3K9 KMTs (G9a and Suv39H1) assembly requires intact DNA. Co-IPs from cells expressing G9a and ORCA or Suv39H1 and ORCA were carried out in the presence or absence of ethidium bromide (EtBr). EtBr selectively inhibits DNA-dependent protein interactions (*Lai and Herr, 1992*). ORCA continued to show interaction with G9a as well as Suv39H1 even in the presence of EtBr (*Figure 1Da,Db*), indicating that these interactions were DNA-independent. The DNA-independent interactions were also corroborated by co-IP experiments in the presence of the nuclease benzonase (data not shown). Furthermore, the interaction of ORCA with G9a as well as Suv39H1 was direct and independent of DNA, as evident by the direct interaction of purified ORCA with G9a/Suv39H1 proteins (*Figure 1Ea,Eb*).

### The association of ORCA and H3K9 KMTs occurs on condensed chromatin

Recent studies have demonstrated that in addition to Suv39H1, G9a/GLP may participate in the establishment of pericentric heterochromatin (*Vassen et al., 2006*; *Dong et al., 2008*; *Kondo et al., 2008*).

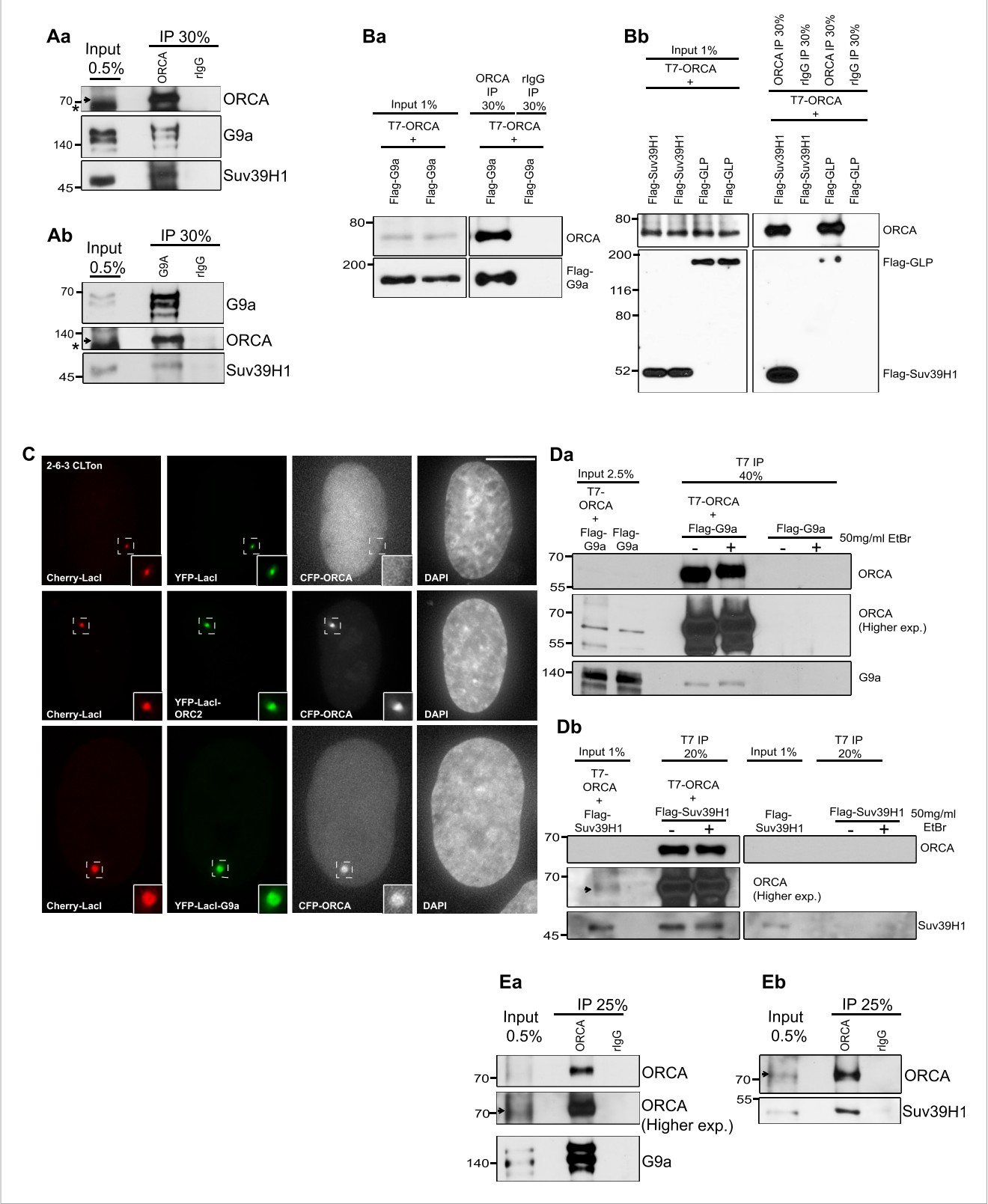

**Figure 1**. ORCA interacts with multiple repressive histone lysine methyltransferases. (**A**) **a** IP using origin recognition complex-associated (ORCA) Ab from U2OS cells. ORCA, G9a, and Suv39H1 were analyzed by immunoblotting (IB). **b** IP using G9a Ab from U2OS cells. Endogenous ORCA, G9a, and Suv39H1 were analyzed by IB. (**B**) **a** and **b** Immunoprecipitation (IP) using ORCA antibody (Ab) from cells expressing T7-ORCA and different Flag-KMTs: **a** H3K9

*Figure 1. continued on next page*

Figure 1. Continued

KMTs G9a; **b** H3K9 KMT GLP and Suv39H1. (**C**) U2OS 2-6-3 CLTon cells co-transfected with individual YFP-LacI-KMTs and CFP-ORCA. Inset represents 150% magnification of the boxed region. (**D**) IP using T7 ab from cells co-expressing T7-ORCA and; **a** Flag-G9a or **b** Flag-Suv39H1 in the presence (+) or absence (–) of EtBr. (**E**) Direct interaction of ORCA and **a** G9a or **b** SUV39H1 using purified proteins. '*' denotes cross reacting band and '➤'denotes ORCA.

The following figure supplement is available for figure 1:

**Figure supplement 1**. Interaction of ORCA with histone methyltransferases.

Since ORCA is enriched at heterochromatic regions, we carried out detailed functional characterization of the interaction of ORCA with Suv39H1 and also with G9a in order to dissect the biological relevance of these associations.

To map the interaction domains of ORCA with G9a and Suv39H1, we generated several truncation mutants of ORCA (*Figure 2Aa*), G9a (*Figure 2Ab*), and Suv39H1 (*Figure 2Ac*). Using co-IP experiments, we observe that the WD repeats of ORCA (truncation mutants 128–647 and 270–647 aa) interacted with G9a (*Figure 2—figure supplement 1Aa*) and Suv39H1 (*Figure 2—figure supplement 1Ab*). We found that the deletion of any one of the WD domains in ORCA resulted in loss of binding to heterochromatin consistent with the fact that the intact β-propeller structure of WD is crucial to maintain its functionality. We also observed that the leucine-rich repeats (LRR)-containing fragment of ORCA (1–127 aa), but not the one containing the linker (1–270 aa), interacted with G9a (*Figure 2—figure supplement 1Aa*) but not with Suv39H1 (*Figure 2—figure supplement 1Ab*). Co-IP experiments demonstrated that the ankyrin repeat (619–965 aa) of G9a (*Figure 2Ba*) and the SET domain (151–412 aa) of Suv39H1 were necessary for interaction with ORCA (*Figure 2Bb*).

To address if the chromatin context affected the interaction between ORCA and G9a, we used the CLTon cells and examined the interaction of YFP-LacI-fused full-length and various truncation mutants of ORCA tethered to the heterochromatic locus, with full-length or truncation mutants of G9a (*Figure 2Ca–Db*, *Figure 2—figure supplement 1Ba,Bb*). The WD domain of ORCA was able to recruit CFP-G9a (*Figure 2Ca,Cb*) corroborating our IP results (*Figure 2—figure supplement 1Aa*). Interestingly, when CFP-LacI-ORCA was co-transfected with YFP-G9a mutants (*Figure 2—figure supplement 1Ba*), not only did the mutant YFP-G9a-1-618, which lacks the ankyrin repeats, show significantly reduced interaction but also YFP-G9a-1-965, which has an intact ankyrin repeat but lacks the SET domain, showed significantly reduced association with the locus (*Figure 2—figure supplement 1Ba,Bb*). We next addressed if the interaction of ORCA with G9a at heterochromatic regions required the catalytic domain of G9a in addition to its ankyrin repeats. YFP-LacI-G9a triple fusion protein was found to be enzymatically active as is evident by the accumulation of H3K9me2 at the CLTon locus upon the tethering of G9a full-length construct (*Figure 2—figure supplement 1C*). Tethering of YFP-LacI-G9a-ΔSET or YFP-LacI-G9a-H1166K (a point mutant, which abolishes the catalytic ability of G9a) to the locus (*Figure 2Da*) failed to recruit CFP-ORCA (*Figure 2Da,Db*). However, co-IP experiments demonstrated that T7-ORCA could interact with GFP-G9a-ΔSET (*Figure 2—figure supplement 1D*). These data suggest that while the ankyrin repeat of G9a is sufficient for the association with ORCA (*Figure 2Ba*), the interaction of ORCA and G9a at the heterochromatin also requires the methylating ability of G9a. Similarly, the interaction between ORCA and Suv39H1 requires the SET catalytic domain (*Figure 2Bb*).

Since we observed the interaction between ORCA and the H3K9 KMTs at the heterochromatic locus, we next asked if the interaction occurred in a chromatin context-dependent manner. We tethered ORCA to the CLTon locus and examined the recruitment of G9a upon induction of transcription from the decondensed locus (*Figure 2Ea*). In the absence of DOX, ~80% of cells showed CFP-G9a recruitment to the locus when YFP-LacI-ORCA was tethered (*Figure 2Ea,Eb*). Upon transcriptional activation, there was a striking reduction (~10%) in CFP-G9a association in the YFP-LacI-ORCA-tethered decondensed locus (*Figure 2Ea,Eb*). ORCA-tethered decondensed locus, in addition to G9a, also failed to recruit HP1α, but contained Cdk9, a component of the pTEFB kinase complex, which is part of the transcription elongation complex (*Figure 2—figure supplement 1E*) Two components of the ORC, Orc2, and Orc3 that require each other for their stability associate with one another at both the condensed and the open chromatin (*Figure 2—figure supplement 1F*). These results indicate that while ORCA can interact directly with G9a and Suv39H1 (*Figure 1Ea,Eb*),

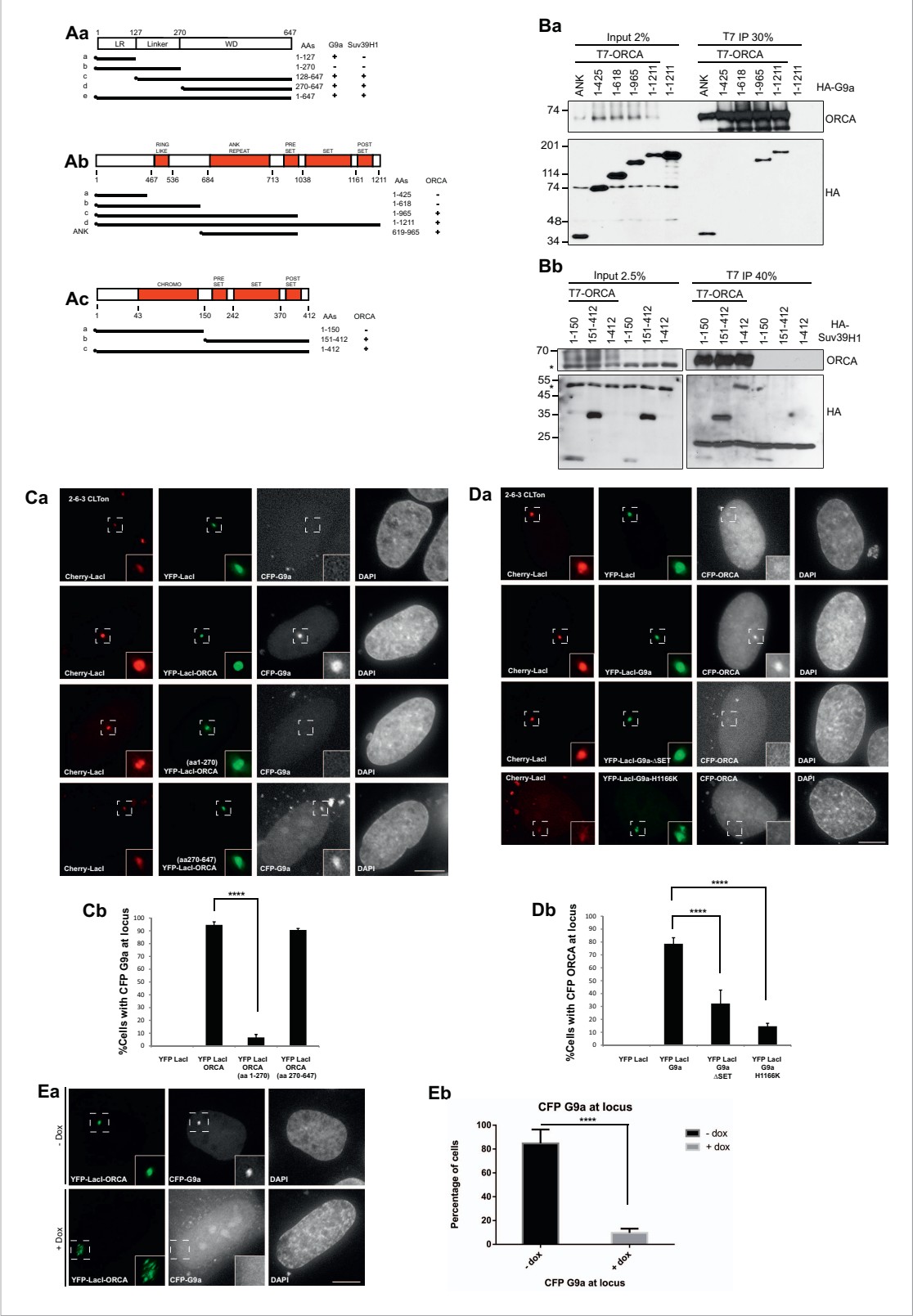

**Figure 2**. ORCA associates with KMTs in a chromatin context-dependent manner. (**A**) **a** Schematic representation of various truncation mutants of ORCA containing a T7-epitope on the N-terminus. The specific domains that can associate with G9a and Suv39H1 based on–IB (*Figure 2—figure supplement 1Aa,Ab*) is depicted as '+'. **b** Schematic representation of various truncation mutants of G9a containing a HA-epitope on the N-terminus. The

*Figure 2. continued on next page*

Figure 2. Continued

interaction domain of G9a that interacts with ORCA (*Figure 2Ba*) is denoted as '+'. **c** Schematic representation of various truncation mutants of Suv39H1 containing a HA-epitope on the N-terminus. The interaction domain of Suv39H1 that interacts with ORCA (*Figure 2Bb*) is denoted as '+'. (**B**) **a** IP in U2OS cells expressing various HA-G9a mutants and T7-G9a using T7 Ab and analysis by T7 and HA-IB. **b** IP in U2OS cells expressing various HA-Suv39H1 mutants and T7-G9a using T7 Ab and analysis by T7 and HA IB. '*' denotes the cross-reacting band. (**C**) **a** Cells co-transfected with YFP-LacI (negative control) and CFP-G9a or YFP-LacI-ORCA or the truncation mutants along with CFP G9a in CLTon cells. **b** The % of cells with CFP-G9a recruited to the locus is plotted. (**D**) **a** Cells co-transfected with YFP-LacI (negative control) and CFP-ORCA or YFP-LacI-G9a wild type and the mutants, which are catalytically inactive along with CFP-ORCA in CLTon cells. **b** The % of cells with CFP-ORCA recruited to the locus. (**E**) **a** U2OS 2-6-3 CLTon cells co-transfected with YFP-LacI-ORCA and CFP-G9a in the presence and absence of doxycycline. **b** The % of cells with CFP-G9a recruited to the locus in both conditions. Scale bars equal 10 μm. Inset represents 150% magnification of the boxed region. Error bars represent s.d., n = 3. ****p < 0.0001.

The following figure supplement is available for figure 2:

**Figure supplement 1**. WD domain of ORCA interacts with H3K9 KMTs.

the interactions could be dependent on or regulated by chromatin within the cells. In a cellular milieu, the fraction of ORCA directly interacting with G9a or Suv39H1, independent of chromatin, is likely to be a very small pool, and therefore, too weak to be detected at the CLTon locus.

## ORCA-ORC and the H3K9 KMTs exist in one single complex

Our earlier work demonstrated the existence of a subset of multiple H3K9 KMTs in a single complex and functional cooperation between these molecules to regulate heterochromatin function and gene expression (*Fritsch et al., 2010*). Since ORCA interacts with different H3K9 KMTs, we investigated if ORCA is an integral component of this multi-KMT complex.

For this purpose, we employed the process of SiMPull analysis (*Figure 3Aa,Ab*) (*Jain et al., 2011*). This method is extremely sensitive and is a tour de force to examine protein complexes and also to accurately calculate the stoichiometry of proteins within the complexes (*Shen et al., 2012*). This approach obviates the need for ensemble experiments that require IPs with large quantities of cell lysates. Our initial estimates predicted that three grams of Flag-HA-Suv39H1-expressing Hela-S3 cell pellet, which is ~3 billion cells, is required for a single glycerol gradient sedimentation to obtain other H3K9 KMT signals detectable by Western blotting (*Fritsch et al., 2010*). However, a relatively higher sensitivity can be achieved by the SiMPull approach by using only a million cells. We first measured the stoichiometry of ORCA bound ORC and H3K9 KMTs, respectively. We used cells co-expressing T7-ORCA and YFP-ORC1 (*Figure 3Ba–Bd*) or T7-ORCA and YFP-KMTs to perform SiMPull (*Figure 3Ca–Dd*). ORCA complexes containing YFP-ORC1 or YFP-KMTs were visualized as isolated fluorescent spots by single-molecule total internal reflection fluorescence (TIRF) microscopy (*Figure 3Bb,Cb,Db*). Pulldown by a control antibody (anti-HA) showed very low non-specific level of fluorescence, thereby demonstrating the high specificity of SiMPull assay. Individual fluorescence spots showed single- or multi-step decreases in fluorescence intensity corresponding to photo-bleaching of individual YFP molecules in a single complex (a representative schematic of the photobleaching analysis is shown in *Figure 3Aa*). After photobleaching analysis of many co-immunoprecipitated YFP-ORC1, YFP-G9a, and YFP-Suv39H1, we found that primarily one molecule each of ORC1, G9a, and Suv39H1 interacts with a single molecule of ORCA (*Figure 3Bd,Cd,Dd*). In addition, we also observed that in a small population, ORCA associates with two molecules of G9a (*Figure 3Cd,Ce*), suggesting that the ORCA-interacting-G9a may also be present as a homodimer.

Next, we investigated whether ORCA bound to KMTs also contains ORC by performing SiMPull using biotin-conjugated anti-T7 antibody with cells co-transfected with YFP-ORC1, mCherry-G9a, and T7-ORCA (*Figure 3Ea*). To mimic endogenous expression levels of these proteins, we first systematically titrated the levels of plasmid transfected to obtain an expression of the candidate protein that is similar to endogenous levels (*Figure 3—figure supplement 1Aa*). Based on this analysis, we transfected $2 \times 10^6$ cells with 100 ng of each plasmid and then carried out SiMPull (*Figure 3—figure supplement 1Aa*, lane3). Complexes of T7-ORCA that contain YFP-Orc1 were detected in the green imaging channel and those containing mCherry-G9a were detected in the red imaging channel (*Figure 3Eb,Ec*). After overlaying the two channels, 39 ± 5% of YFP-ORC1 molecules colocalized with mCherry-G9a, indicating that all three proteins, ORC, ORCA, and G9a are found in a substantial fraction of single complexes

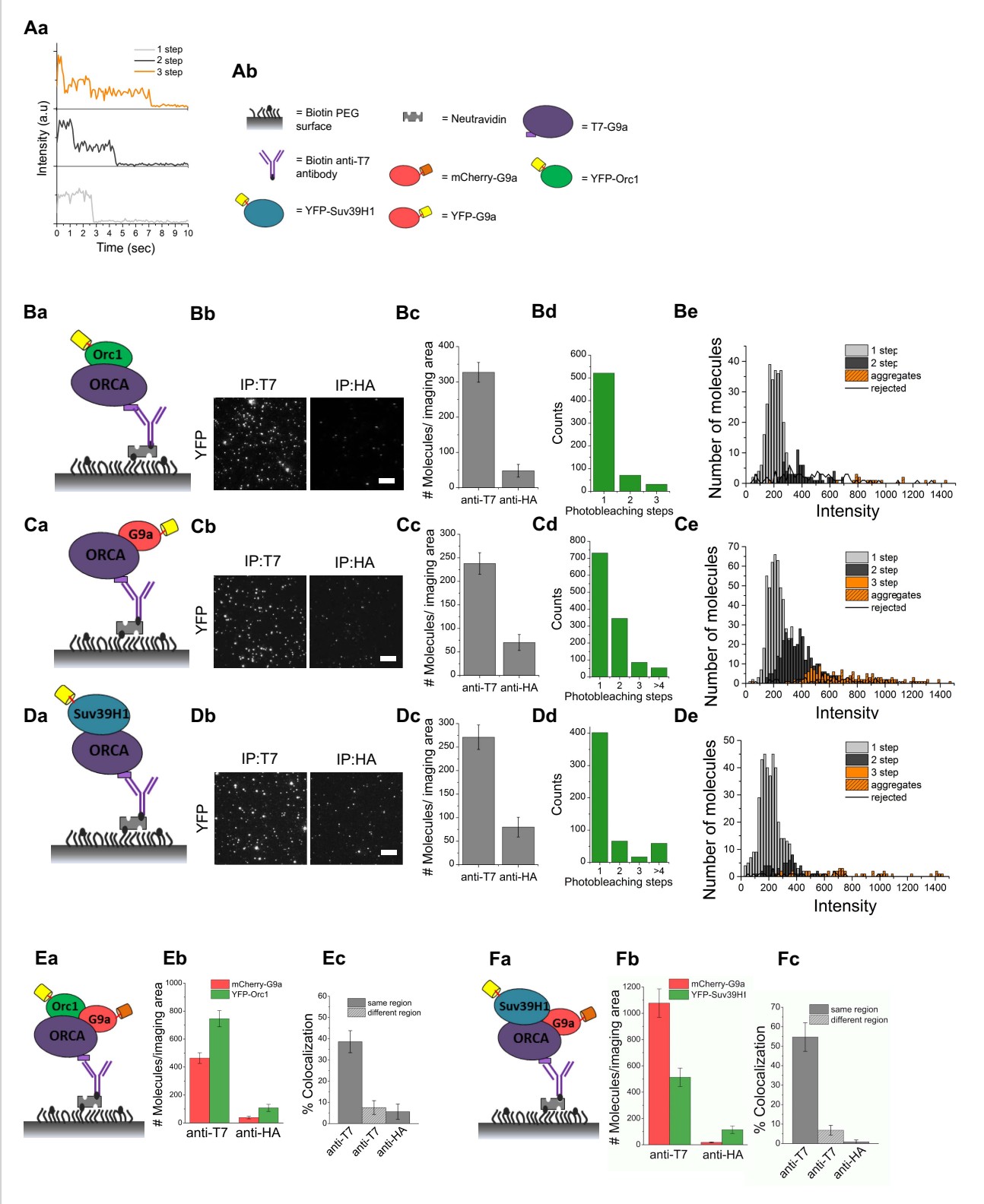

Figure 3. ORCA and H3K9 KMTs exist in one multimeric complex. (**A**) **a** Representative single-molecule fluorescence time trajectories for YFP-tagged molecules that exhibit one-step, two-step, and three-step photobleaching. **b** Key to the schematics of the SiMPull assay. (**B**) **a** and **b** Schematic and total internal reflection fluorescence (TIRF) images of YFP molecules pulled down from U2OS cell lysates expressing T7-ORCA and YFP-ORC1 using biotinylated T7 Ab. The same lysate incubated with biotinylated HA Ab served as the control. **c** Average number of YFP fluorescent molecules per imaging
*Figure 3. continued on next page*

Figure 3. Continued

area (5000 µm²). **d** Photobleaching step distribution for YFP-ORC1 bound to T7-ORCA. Note 1:1 ratio of ORCA to Orc1. **e** Intensity profiles of the YFP-ORC1 molecules bound to T7-ORCA. (**C**) **a–d** ORCA-G9a pulldown. Shown are YFP molecules pulled down from U2OS cell lysates expressing T7-ORCA and YFP-G9a. Note 1:1 or 1:2 ratio of ORCA to G9a. **e** Intensity profiles of YFP-G9a molecules bound to T7-ORCA. (**D**) **a–d** ORCA-Suv39H1 pulldown. Shown are YFP molecules pulled down from U2OS cell lysates expressing T7-ORCA and YFP-Suv39H1. Note 1:1 ratio of ORCA to Suv39H1. **e** Intensity profiles of YFP-Suv39H1 molecules bound to T7-ORCA. (**E**) **a–c** Determination of ORCA complexes containing both ORC and G9a by SiMPull and colocalization analyses. **a** Schematic of YFP and mCherry molecules pulled down from U2OS cell lysates expressing T7-ORCA, YFP-ORC1, and mCherry-G9a using biotinylated T7 Ab. The same lysate incubated with biotinylated HA Ab served as the control. **b** Average number of YFP and mCherry fluorescent molecules per imaging area (5000 µm²). **c** Note 39 ± 5% overlap. Transfection condition used as indicated in *Figure 3—figure supplement 1Aa*, lane3. (**F**) **a–c** Determination of ORCA complexes containing multiple H3K9 KMTs by SiMPull and colocalization analyses. **a** Schematic of YFP and mCherry molecules pulled down from U2OS cell lysates expressing T7-ORCA, YFP-Suv39H1, and mCherry-G9a using biotinylated T7 Ab. The same lysate incubated with biotinylated HA Ab served as the control. **b** Average number of YFP and mCherry fluorescent molecules per imaging area (5000 µm²). **c** Note 55 ± 7% colocalization. Transfection condition used as indicated in *Figure 3—figure supplement 1Ba*, lane3. Scale bars, 10 µm. Error bars represent s.d., n = 3.

The following figure supplement is available for figure 3:

**Figure supplement 1**. ORC-ORCA-H3K9 KMTs exist in a single complex.

(*Figure 3Ec*). SimPull from cells that were transfected with higher concentration of plasmid showed similar extent of co-localization (*Figure 3—figure supplement 1Ab,Ac*). The results are consistent with our earlier study showing that ORCA is protected from ubiquitin-mediated proteolysis when bound to ORC and as a result is always associated with ORC (*Shen and Prasanth, 2012*). Next, we tested whether multiple H3K9 KMTs exist in a single complex with ORCA using triple transfections of T7-ORCA, YFP-Suv39H1, and mCherry-G9a in U2OS cells (*Figure 3Fa,Fc*, plasmid titrations: *Figure 3—figure supplement 1Ba*, lane3 used for the experiment). Interestingly, we could observe ~55 ± 7% of YFP-Suv39H1 colocalized with mCherry-G9a, (*Figure 3Fc*). Similar results were obtained with higher concentration of plasmid transfection: (*Figure 3—figure supplement 1Bb,Bc*), suggesting the existence of a significant amount of ORCA-G9a-Suv39H1 complex. The true degree of cohabitation may be even higher because the fluorescent proteins may not all mature into active chromophores. This leads to dark molecules and appearance of either only green or only red spots even though both the KMTs are present in a complex. In addition, unequal expression of the transfected KMTs or the presence of endogenous KMTs in the complexes may also lead to a reduction in cohabitation detected by SiMPull. Finally, only a subset of G9a and Suv39H1 may exist as a single complex with ORCA (similar to reported data [*Fritsch et al., 2010*]). Elucidation of three different proteins in a single complex is one of the promised capabilities of SiMPull (*Jain et al., 2011*), and the data we present here constitute one of the first demonstrations of such a capability.

In order to corroborate our SimPull observations on the existence of ORC-ORCA-H3K9 KMTs and G9a-ORCA-Suv39H1 in a single complex, we utilized sequential IPs. We carried out triple transfections of T7-ORCA, HA-ORC1, and Flag-G9a in U2OS cells, followed by immunoprecipitation of HA-ORC1. Following HA peptide elution, the eluate was used for T7-Ab immunoprecipitation. T7-ORCA was immunoprecipitated, and a robust co-IP of Flag-G9a was detected (*Figure 3—figure supplement 1C*). This further confirmed the existence of ORC-ORCA-H3K9 KMTs in a single complex. Similarly, we performed triple transfections of T7-ORCA, HA-G9a, and Flag-Suv39H1 in U2OS cells followed by immunoprecipitation of HA-G9a. Following HA peptide elution, the eluate was used for T7-Ab immunoprecipitation. T7-ORCA was immunoprecipitated, and a robust co-IP of Flag-Suv39H1 was detected (*Figure 3—figure supplement 1D*), further confirming the existence of multiple H3K9 KMTs in a single complex with ORCA. The exogenous expression of Suv39H1 did not affect the association of G9a and ORCA; similarly, the exogenous expression of G9a did not compromise the association of Suv39H1 and ORCA (*Figure 1—figure supplement 1Aa,Ab*).

## Loss of ORCA causes global changes in H3K9-containing heterochromatin structure

Previous studies indicated that ORCA along with ORC associates with heterochromatic regions (*Prasanth et al., 2010*; *Shen et al., 2010*) and also specifically binds to repressive histone and DNA marks (*Vermeulen et al., 2010*). Since ORCA interacts with both H3K9me2 and H3K9me3-catalyzing

enzymes, we examined the direct binding of ORCA to these marks. We performed peptide pull-downs with N-terminal peptides of histone H3 with the K9 differentially modified with acetylation or mono-, di-, or tri-methylation (*Figure 4—figure supplement 1A*). Iodoalkyl agarose-conjugated peptides were incubated with purified His-tagged-ORCA. We found that ORCA displayed increased interaction with mono-, di-, and tri-methylated H3K9 compared to unmodified or K9-acetylated H3 peptides (*Figure 4—figure supplement 1A*).

To get a more quantitative estimation of the affinity of ORCA for differentially methylated H3K9, we have employed SiMPull for the first time as a potential substitute for isothermal calorimetry. Biotinylated histone H3 N-terminal tails were immobilized on passivated quartz slides followed by passing lysates expressing full-length YFP-ORCA or the fragment 1–127 aa, which contains only the LRR and lacks WD domain necessary for binding to methylated histones (*Figure 4Aa*). The level of

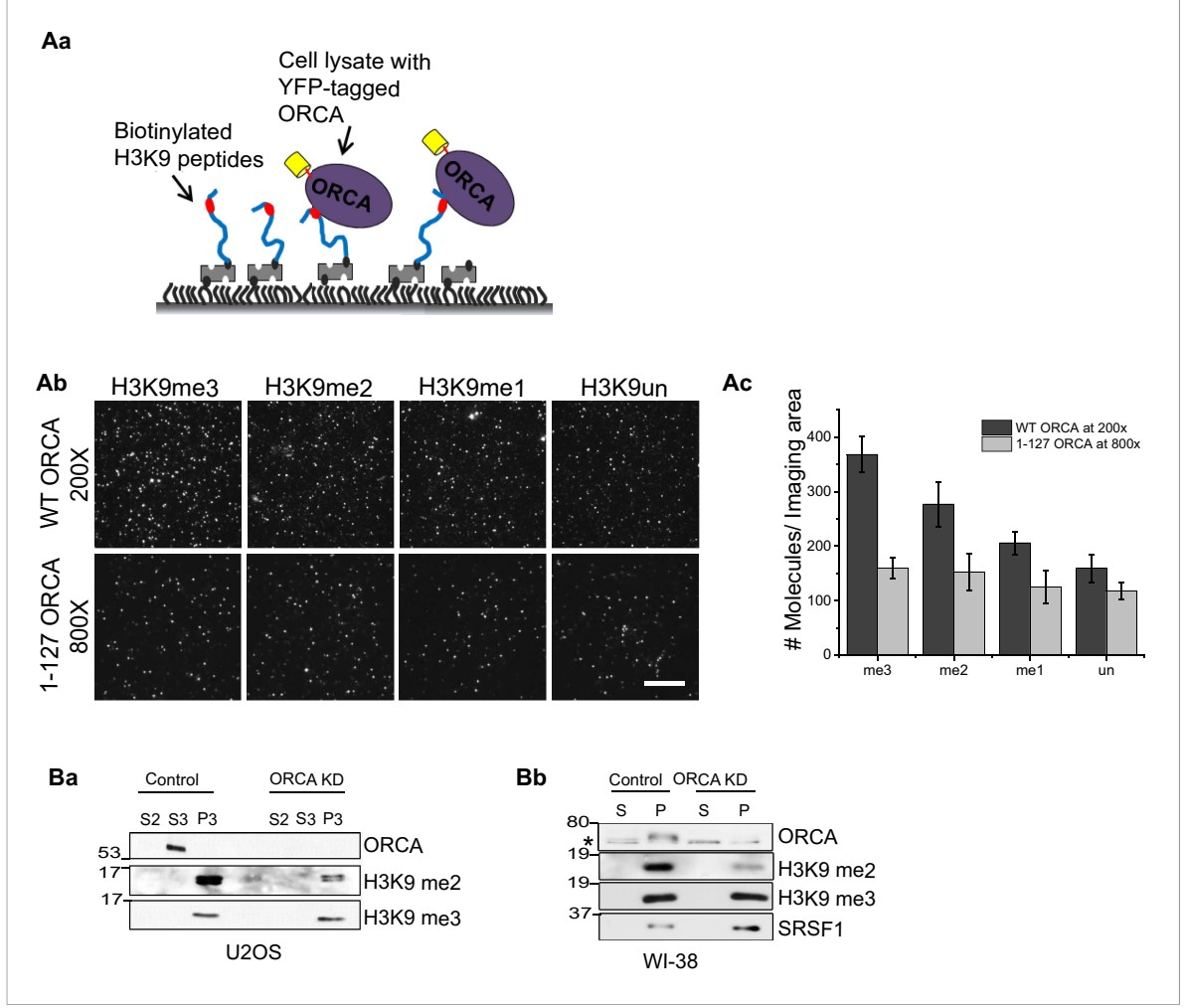

**Figure 4**. ORCA binds and regulates levels of H3K9 methylation. (**A**) **a** Schematic of experimental setup for peptide pulldown and analyses by SiMPull. **b** TIRF images of YFP-ORCA WT and 1–127 aa pulled down by H3K9 modified peptides. Note that the YFP-ORCA WT and 1–127 aa truncation mutant expressing lysates were diluted so that the concentration of the overexpressed proteins is comparable (200 and 800 times, respectively for WT and 1–127 aa). **c** Average number of fluorescent molecules per imaging area. Scale bars, 10 µm. (**B**) **a** Chromatin fractionation in ORCA-depleted U2OS cells followed by IB analysis of H3K9me2 and me3. **b** Chromatin fractionation in ORCA-depleted diploid fibroblasts, WI38 followed by IB analysis of H3K9me2 and me3. Splicing factor, SRSF1 is shown as a loading control. Error bars represent s.d., n = 3. S and S2-cytosolic; S3-nuclear soluble and MNase sensitive; P: nuclear; P3: nuclear insoluble and MNase resistant fraction.

The following figure supplement is available for figure 4:

**Figure supplement 1**. ORCA binding to H3K9 peptides.

YFP-ORCA expression was quantitated by a direct anti-GFP pull-down with both the lysates (*Figure 4—figure supplement 1Ba*) and analyses of the average number of molecules pulled down (*Figure 4—figure supplement 1Bb,Bc*). The lysates were then diluted so that the expression of the YFP-tagged proteins is nearly equal and passed through the flow chambers containing the immobilized peptides. Analysis of the average number of molecules pulled down by the peptides revealed that ORCA has the highest affinity for H3K9me3 followed by for me2 and me1 (*Figure 4Ab,Ac*). YFP-ORCA 1–127 aa showed a low-basal binding to all the peptides corroborating the necessity of WD domain of ORCA for specifically recognizing methylated histones.

To further determine whether ORCA is required for the establishment of these histone marks on chromatin, depletion of ORCA (siRNA-mediated knockdowns [KDs]) was carried out both in cancerous cells (U2OS) and diploid fibroblasts (WI-38), and the total levels of H3K9me2 and H3K9me3 were analyzed by immunoblotting (IB) (*Figure 4Ba,Bb*). In both the cell lines, upon depletion of ORCA, the levels of H3K9me2 decreased while H3K9me3 remained unchanged at the resolution of Western blotting (*Figure 4Ba,Bb*).

To determine the involvement of ORCA in the genome-wide status of H3K9 methylation, we conducted H3K9me3 ChIP-sequencing upon ORCA depletion. We observed a significant decrease in H3K9me3 ChIP-seq signal upon ORCA KD (*Figure 5A,B*). Around 18% of the detected peaks showed highly significant (more than fivefold decrease) changes in H3K9 tri-methylation in cells lacking ORCA (*Figure 5B*, *Supplementary file 1* and total H3K9 in *Figure 5—figure supplement 1Aa*). Interestingly, several regions did not show significant change in H3K9me3 association upon ORCA KD (*Figure 5—figure supplement 1Ab*). Since ORCA associates with heterochromatin, we specifically analyzed the number of reads of satellite repeats in H3K9me3 ChIP and found that there was a significant reduction of this mark at these regions of the genome upon ORCA KD (*Figure 5Ca*). Furthermore, H3K9me3 showed less association both with the telomeric (TAR1) and centromeric (SST1) repetitive DNA in cells lacking ORCA (*Figure 5Cb,Cc*).

Our attempts on H3K9me2 ChIP-seq did not succeed because of the technical challenge associated with sequencing the broad H3K9me2 peaks. Similar problems with H3K9me2 ChIP-seq have been previously reported by other studies (*Yuan et al., 2009*). As an alternate, regions that showed significant reduction of H3K9me3 in the ChIP-seq experiment (as evident by the wiggle plots; *Figure 5Da–Dd* and *Figure 5—figure supplement 1Ba,Bb*) were chosen for H3K9me2 ChIP-qPCR (quantitative PCR) validation (*Supplementary file 2*). These regions also consistently showed a significant reduction of H3K9me2, corroborating the decrease seen in Western blotting (*Figure 5—figure supplement 1Ca, Cb*). *C-FOS*, a gene that does not associate with these repressive histone marks, was used as a negative control (*Figure 5—figure supplement 1Bc–Cb*). To determine whether the H3K9-targets are directly regulated by ORCA, we conducted ChIP using HA antibody in HA-ORCA expressing stable cell line. This allowed us to address if ORCA is associated with H3K9-occupied genomic sites. We observed a strong enrichment of ORCA at the H3K9-enriched loci (*Figure 5Ea*), while ORCA binding to *C-FOS*, a region devoid of H3K9, was comparable to that of IgG (*Figure 5Eb*).

To understand the mechanism of reduction of H3K9 methyl marks upon ORCA depletion, we first determined whether the protein stability or chromatin association of G9a and Suv39H1 was altered upon ORCA loss. Our data revealed that the total cellular levels of G9a and Suv39H1 were not reduced upon ORCA KD (*Figure 5—figure supplement 1D*). We next addressed if the loading of these KMTs onto chromatin is impaired upon ORCA KD. To investigate this, we performed G9a and Suv39H1 ChIP upon ORCA KD and analyzed the association of these enzymes to the loci that show H3K9me2 and H3K9me3 reductions. Suv39H1 showed a decrease at these loci (*Figure 5Fa,Fb*), indicating that the loading of Suv39H1 to these regions is reduced upon ORCA depletion. G9a association with these regions showed either no alteration or an increase at some regions (*Figure 5—figure supplement 1Ea,Eb*), indicating that the reduction in H3K9me2 that was observed was possibly due to impaired catalytic activity of G9a.

## ORCA stabilizes the multimeric H3K9 KMT complex

Next, we addressed if ORCA facilitates the assembly of the multimeric H3K9 KMT megacomplex. To address this, the association between the components of the KMT megacomplex, namely G9a and Suv39H1, was evaluated in cells that were treated with control or ORCA siRNAs. Flag-G9a and HA-Suv39H1 were expressed, and HA IP was carried out in control and ORCA-depleted (ORCA KD) cells. ORCA KD showed close to 50% reduction of Suv39H1 that co-immunoprecipitated with G9a

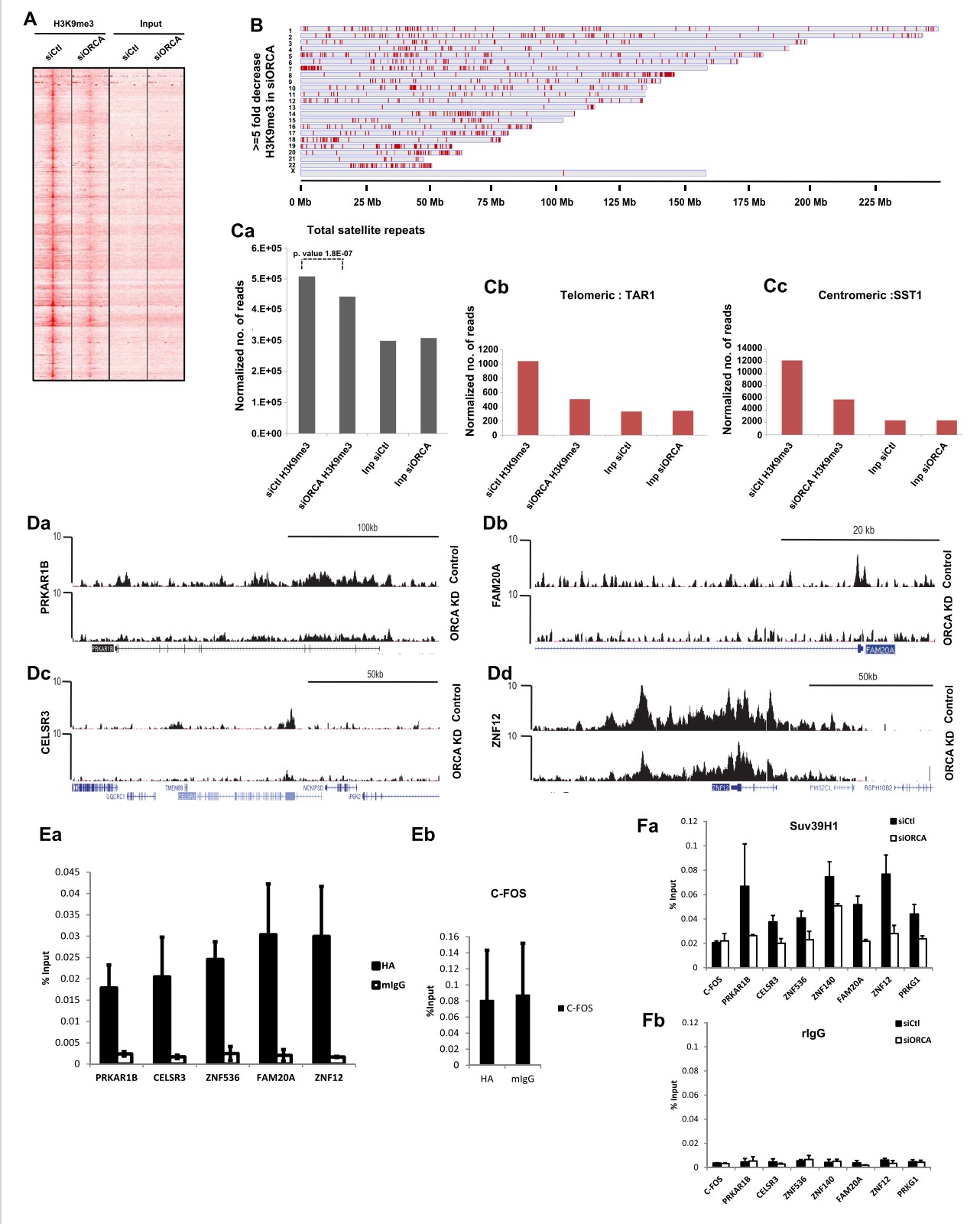

**Figure 5**. Loss of ORCA leads to significant reduction in H3K9 methylation. (**A**) Model-based analysis of ChIP-sequencing (MACS) 1.4 peaks analysis of H3K9me3 ChIP-seq in control and ORCA-depleted cells. (**B**) Regions showing greater than fivefold decrease in H3K9me3 upon ORCA knockdown (KD) plotted along the length of the chromosomes in which they reside. (**C**) **a** Normalized number of reads of repetitive sequences in control and ORCA KD

*Figure 5. continued on next page*

*Figure 5. Continued*

H3K9me3 ChIP-seq. Normalized number of reads of-**b** Telomeric repetitive sequences and **c** Centromeric repetitive sequences in control and ORCA KD H3K9me3 ChIP-seq. (**D**) **a**–**d** Representative regions showing significant decrease in reads in H3K9me3 ChIP on ORCA KD compared to the control. (**E**) **a** HA-ORCA ChIP at H3K9me3-target sites and (**b**) C-FOS. (**F**) **a** Suv39H1 ChIP and **b** IgG ChIP at regions showing decrease in H3K9me3. Error bars represent s.d., n = 3. C-FOS is shown as negative control.

The following figure supplement is available for figure 5:

**Figure supplement 1**. ORCA depletion causes changes in chromatin architecture.

(*Figure 6Aa,Ab*). This observation suggested that the stability of the KMT complex requires ORCA. We used SiMPull to obtain an accurate quantitative estimate of the reduction (*Figure 6Ba–Bc*). YFP-Suv39H1 and mCherry-G9a were expressed in cells depleted of ORCA. GFP pull-down was carried out, and the number of mCherry-G9a molecules associated with Suv39H1 was calculated (*Figure 6Bb,Bc*). ORCA KD led to ~50% reduction in the complexes containing YFP-Suv39H1 and mCherry-G9a (note, 24 ± 3% mCherry-G9a pulled down by YFP-Suv39H1 in control vs 15 ± 1% in ORCA knock-down cells; *Figure 6Bc*). These results support the argument that ORCA acts as a scaffold protein that is necessary for stabilizing a subset of the complexes containing multiple H3K9 KMTs.

To further confirm the role of ORCA as a scaffold protein, we addressed if over-expressing ORCA leads to any increase in G9a and Suv39H1-containing complexes. We performed triple transfections of YFP-Suv39H1, mCherry-G9a, and T7-ORCA. YFP-Suv39H1 SimPull was carried out, and the number of mCherry-G9a molecules pulled down was analyzed as a percentage of YFP-Suv39H1 pull-down (*Figure 6Ca,Cc*). The presence of full-length T7-ORCA showed 25 ± 1% association between Suv39H1 and G9A. The mutant T7-ORCA (1–270) that does not interact with either G9a or Suv39H1, when expressed along with G9a and Suv39H1 showed 14 ± 3% of mCherry-G9a in complex with YFP-Suv39H1. By contrast, the other T7-ORCA mutant (128–647) that interacts with both G9a and Suv39H1 stabilized mCherry-G9a and YFP-Suv39H1 complexes in the cell (29 ± 6%; *Figure 6Cb,Cc*).

These results collectively indicated that ORCA, by acting as a scaffold protein, stabilizes the association of multiple KMTs in a single complex. In the absence of ORCA, the integrity of this complex is compromised, leading to the reduction in the KMT-associated enzymatic activity and a subsequent reduction of H3K9me2 and H3K9me3 patterns on chromatin.

## Changes in chromatin organization upon ORCA loss affect heterochromatin replication

In general, chromatin at the nuclear periphery is significantly enriched with H3K9me2, whereas H3K9me3 is preferentially enriched around nucleolus (*Yokochi et al., 2009*). Typically, both of these regions replicate late during S-phase indicating that in general repressive histone marks-containing differentially condensed chromatin influences replication timing and chromatin positioning (*Julienne et al., 2013*). Therefore, we investigated whether the changes in H3K9me2 and H3K9me3 deposition in specific chromatin regions, upon ORCA depletion, also influences their replication timing. We depleted ORCA in U2OS cells and then synchronized the cells so as to analyze the spatio-temporal regulation of replication during S-phase (*Figure 7A*). Samples were collected at 4, 8, 12 hr post release from aphidicolin arrest with BrdU pulse-labeling prior to sample collection. This was followed by immunofluorescence to score for cells in early, mid, and late S-phase of cell cycle (*Figure 7—figure supplement 1A*). At 8 hr and 12 hr time points post-aphidicolin release, ORCA depletion caused dramatic reduction in cells showing late replication patterns (*Figure 7B*). BrdU (Bromodeoxyuridine)-PI flow cytometry results showed a significant reduction in BrdU incorporation in ORCA-depleted cells without significant changes in the early S-phase (*Figure 7C*). To determine if the changes in late replication pattern are a reflection of changes in the heterochromatin organization, we examined the replication timing of regions that showed a reduction in H3K9me2 and H3K9me3 upon ORCA KD. Initial analysis of the available repli-seq data set from various human cell lines in UCSC Genome Browser and ENCODE consortium revealed that the replication timing of large domains remains the same across cell lines. We, therefore, compared the H3K9me3 ChIP-seq data set to the HeLa repli-seq data set (*Figure 7D*). HeLa-S3 G1b and HeLa-S3 S1 are deep sequencing data sets for late G1 and

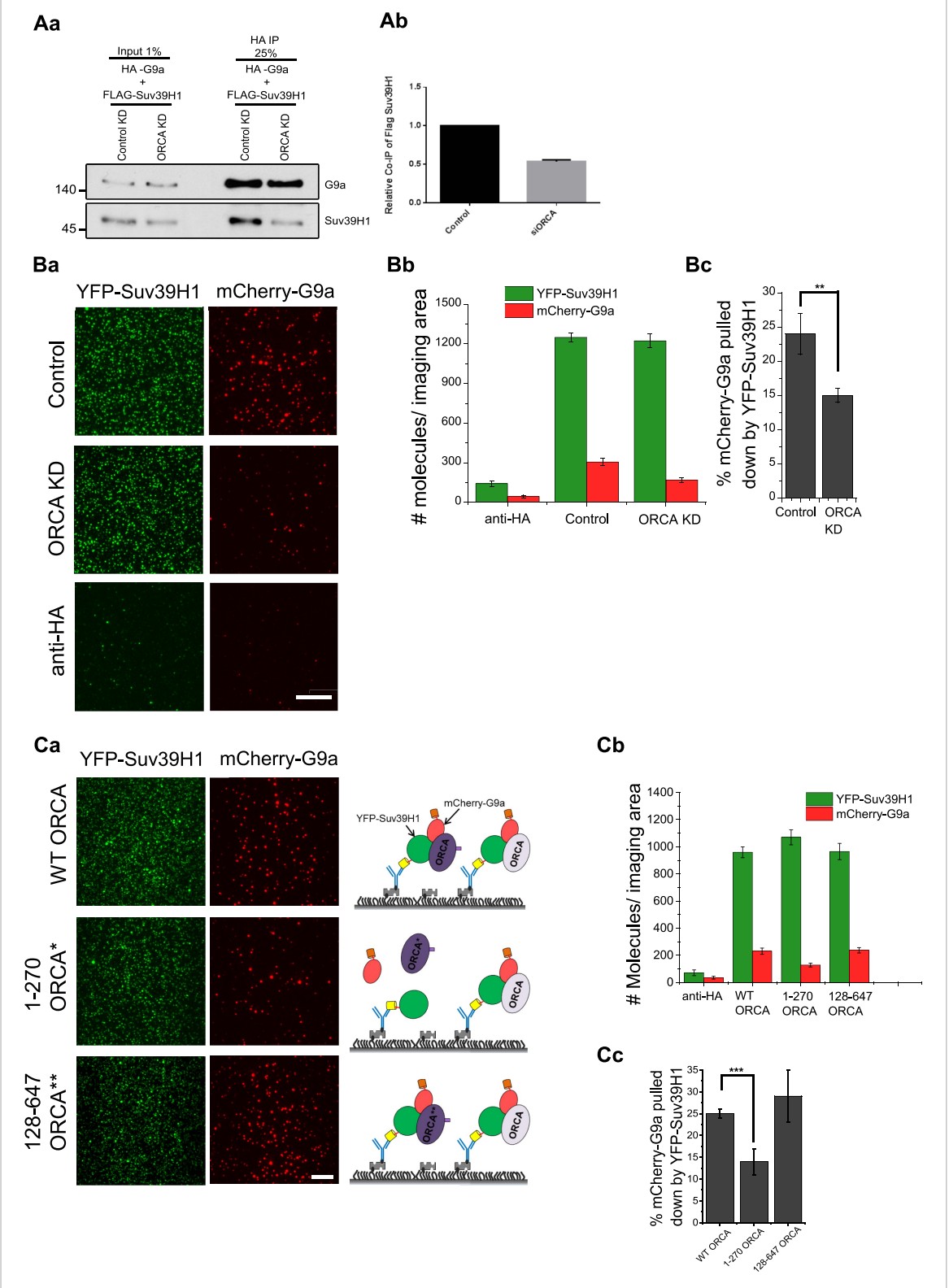

**Figure 6**. ORCA is a scaffold for G9a-Suv39H1 complexes. (**A**) **a**–**b** HA-IP in control and ORCA-depleted U2OS cells co-expressing with HA-G9a and Flag-Suv39H1. (**B**) **a** TIRF images of GFP SiMPull in control and ORCA-depleted U2OS cells co-transfected with YFP-Suv39H1 and mCherry-G9a. The same lysates incubated with biotinylated HA Ab served as the control. **b** Average number of YFP fluorescent molecules per imaging area (5000 µm²). **c** The % of mCherry-G9a pulled down by YFP Suv39H1 in control and ORCA KD. (**C**) **a** TIRF images of GFP SiMPull in U2OS cells transiently transfected with

*Figure 6. continued on next page*

Figure 6. Continued

YFP-Suv39H1, mCherry-G9a, and T7-ORCA full-length or truncation mutant 1–270 or 128–647. The same lysates incubated with biotinylated HA Ab served as the control. **b** Average number of YFP fluorescent molecules per imaging area (5000 μm²). **c** The % of mCherry-G9a pulled down by YFP-Suv39H1. The % of mCherry-G9a pulled down by YFP-Suv39H1 in WT-ORCA is 25 ± 1%; 1–270 ORCA is 14 ± 3%; and 128–647 ORCA is 29 ± 6%. Scale bars, 20 μm. Error bars represent s.d., n = 3. **p < 0.01, ***p < 0.001.

early S replicating regions in HeLa-S3 cells (*Hansen et al., 2010*). The chromosomal regions that are replicating at these two stages are shown in black (early) and late (gray) along the length of the chromosome (*Figure 7D* and *Figure 7—figure supplement 1B*).

Using the data set mentioned above, we examined the replication timing of the regions that showed reduction in H3K9me3 by ChIP-seq. On chromosome 19, the total H3K9me3 peaks in the control sample and the regions, which show greater than fivefold decrease in H3K9me3 upon ORCA depletion are represented as black bars above the HeLa-S3 G1b and HeLa-S3 S1 tracks. Upon ORCA depletion, most of the affected H3K9me3 peaks resided in late-replicating domains (*Figure 7D* and *Figure 7—figure supplement 1B*). This coupled with the loss of late-replication patterns by BrdU IF in ORCA-depleted cells made us hypothesize that ORCA could also regulate the replication of late-replicating regions.

Loss of ORCA could be causing either changes in replication timing of late-replicating regions or the complete loss of replication of these regions. To investigate these possibilities, we conducted BrdU ChIP in control and ORCA-depleted U2OS cells (*Figure 7—figure supplement 1Ca–Cc*). We depleted ORCA and arrested the cells using aphidicolin. The cells were then released into S-phase, pulse-labeled with BrdU, and analyzed by BrdU ChIP at different time points post-release (0, 4, 8, and 12 hr). The replication timing of various loci that showed significant reduction in H3K9me2 and me3

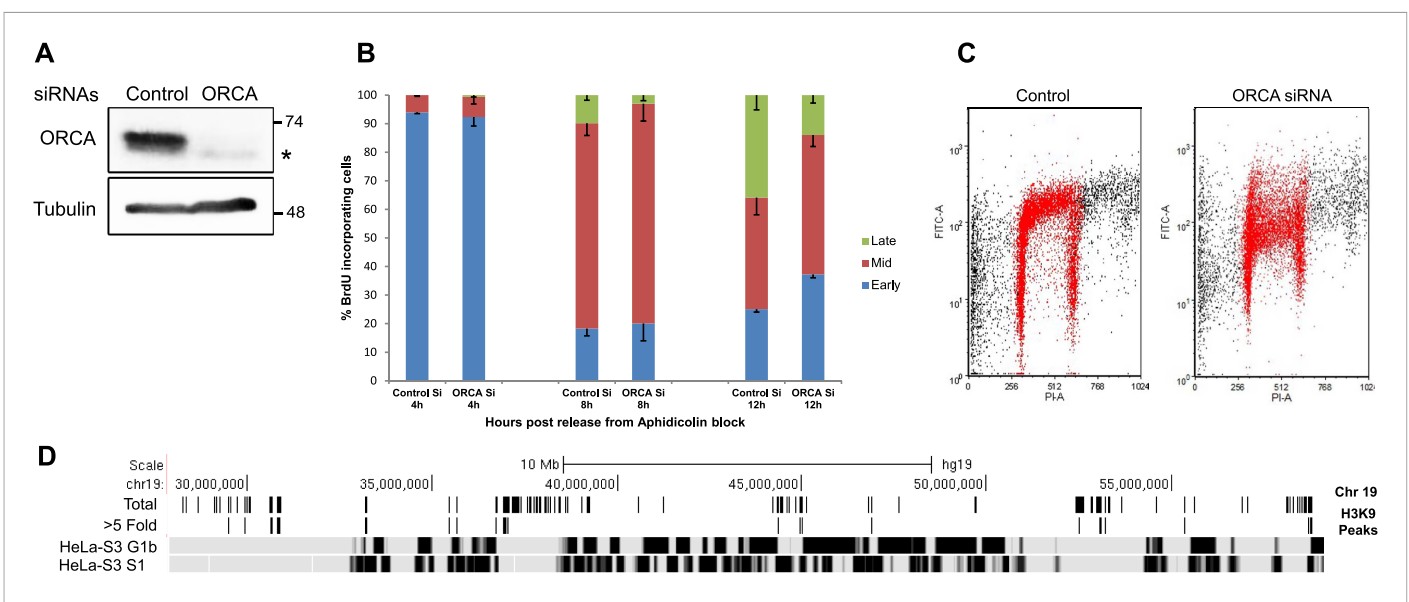

**Figure 7**. Loss of ORCA causes defects in heterochromatin organization. (**A**) IB showing efficient siRNA-mediated KD of ORCA. (**B**) Distribution of S-phase cells displaying early, mid, and late replication patterns in control and ORCA KD cells. Error bars represent s.d., n = 3 independent experiments with 500 BrdU positive cells scored in each. (**C**) BrdU-PI flow cytometry of control and ORCA KD cells. (**D**) Replication timing of genomic regions that show reduced H3K9me3 upon ORCA KD. Gray bars represent late-replicating domains, and black bars denote early replicating domains. HeLa-S3 G1b and HeLa-S3 S1 are late G1 and early S cell cycle fractions that together represent the early replicating regions of the genome.

The following figure supplement is available for figure 7:

**Figure supplement 1**. Depletion of ORCA alters the replication timing.

(*Figure 5D* and *Figure 5—figure supplement 1Ca*) was assessed by quantitative PCR. We observed changes in replication timing of these loci (*Figure 7—figure supplement 1Cb,Cc* shows two representative loci CELSR3 and FAM20A) upon loss of ORCA. For example, in control cells a significant population of CELSR3 locus replicates in late S (12 hr post release) as evident by the BrdU ChIP signal at 12 hr. Upon ORCA KD, there is a significant increase in the population of the locus replicating during early S (4-hr time point) and a concomitant reduction in BrdU ChIP signal at mid and late S (8 and 12 hr time points) (*Figure 7—figure supplement 1Cb*). The replication timing of C-FOS, a region that serves as a control showing no changes in H3K9me2 and me3 upon loss of ORCA, remains unaffected (*Figure 7—figure supplement 1Ca*), suggesting that the replication timing changes observed in ORCA-depleted cells may not be because of the direct role of ORCA/ORC in establishing the pre-replicative complex.

## ORCA's role in heterochromatin organization is independent of its role in preRC assembly

We have previously demonstrated that ORCA plays a key role in replication initiation (*Shen et al., 2012*). We addressed whether the observed defects in heterochromatin organization and replication patterning in cells lacking ORCA are due to defects in preRC assembly or reflect the more direct role of ORCA in heterochromatin organization. While it is well-known that ORC (along with ORCA) associates with heterochromatic regions in post-replicated cells in metazoans (*Prasanth et al., 2004*; *Shen et al., 2010*), its direct role in heterochromatin organization vs heterochromatin replication licensing has remained to be understood.

In order to understand ORCA's role in chromatin organization and if it is independent of its role in preRC function, we wanted to deplete ORCA after the establishment of pre-replication complex (post-G1 phase) but before DNA synthesis began. Depletion of a protein within a short, specific time window by RNA interference is challenging because even if the mRNA levels are dramatically reduced, the protein levels could persist for significantly longer times. This necessitates the use of a strategy that utilizes post-translation degradation process for reducing proteins levels efficiently. To achieve this, we utilized a commercially available Proteotuner kit (Clonetech). Briefly, an siRNA resistant version of T7-ORCA (T7-ORCA-siRNA NTV: non-targetable version) was tagged with a destruction signal or DD (destabilization domain) tag, a destabilization domain of the FKBP protein (*Figure 8A*; [*Banaszynski et al., 2006*]). This signal is recognized by the proteosomal machinery and results in the rapid degradation of the exogenous ORCA. In the presence of a ligand, Shield1, the DD tag is masked by the binding of Shield1, thereby preventing the degradation of the exogenous ORCA protein.

In order to determine whether DD-T7-ORCA-siRNA NTV is functional and can substitute for endogenous ORCA, we examined whether it could efficiently rescue ORC levels on chromatin upon depletion of endogenous ORCA (*Shen et al., 2010*). We transfected DD-T7-ORCA-siRNA NTV into U2OS along with siRNA to KD endogenous ORCA. Following two rounds of siRNA treatment (48 hr) in the presence of DD-T7-ORCA-siRNA NTV (*Figure 8B*), we examined the loading of ORC2 on chromatin and compared it to ORC2 loading in control and ORCA-depleted cells. We observed that while ORC2 loading was affected upon ORCA depletion (levels of ORC2 decrease in the chromatin fraction with concomitant increase in the supernatant fraction), the expression of DD-T7-ORCA-siRNA NTV construct rescued this phenotype by restoring the levels of ORC2 on chromatin to an extent comparable to that of the control (*Figure 8C*). In addition, we also carried out immunoprecipitation of DD-T7-ORCA and found that it efficiently interacts with endogenous ORC2 (*Figure 8—figure supplement 1A*), further confirming that DD-T7-ORCA is functional.

To determine if the role of ORCA in heterochromatin organization is independent of its role in preRC assembly, we transfected U2OS cells with DD-T7-ORCA-siRNA NTV. We then depleted endogenous ORCA by using ORCA siRNA, while the levels of exogenous DD-T7-ORCA-siRNA NTV were maintained by growing the cells in the presence of Shield1. We synchronized the cells at early S by using aphidicolin and then degraded exogenous DD-T7-ORCA at early S by removing shield from the medium. Removal of Shield1 resulted in the loss of exogenous ORCA (in addition to endogenous ORCA that was removed by siRNA treatment [*Figure 8D*]). The cells were then allowed to progress through S-phase and chromatin organization and replication were examined at different time points during S-phase. Specific depletion of ORCA only in post-G1 cells also resulted in reduction in the H3K9me2 levels (*Figure 8D*). This demonstrates that the heterochromatin is disorganized in the absence of ORCA in post-G1 cells.

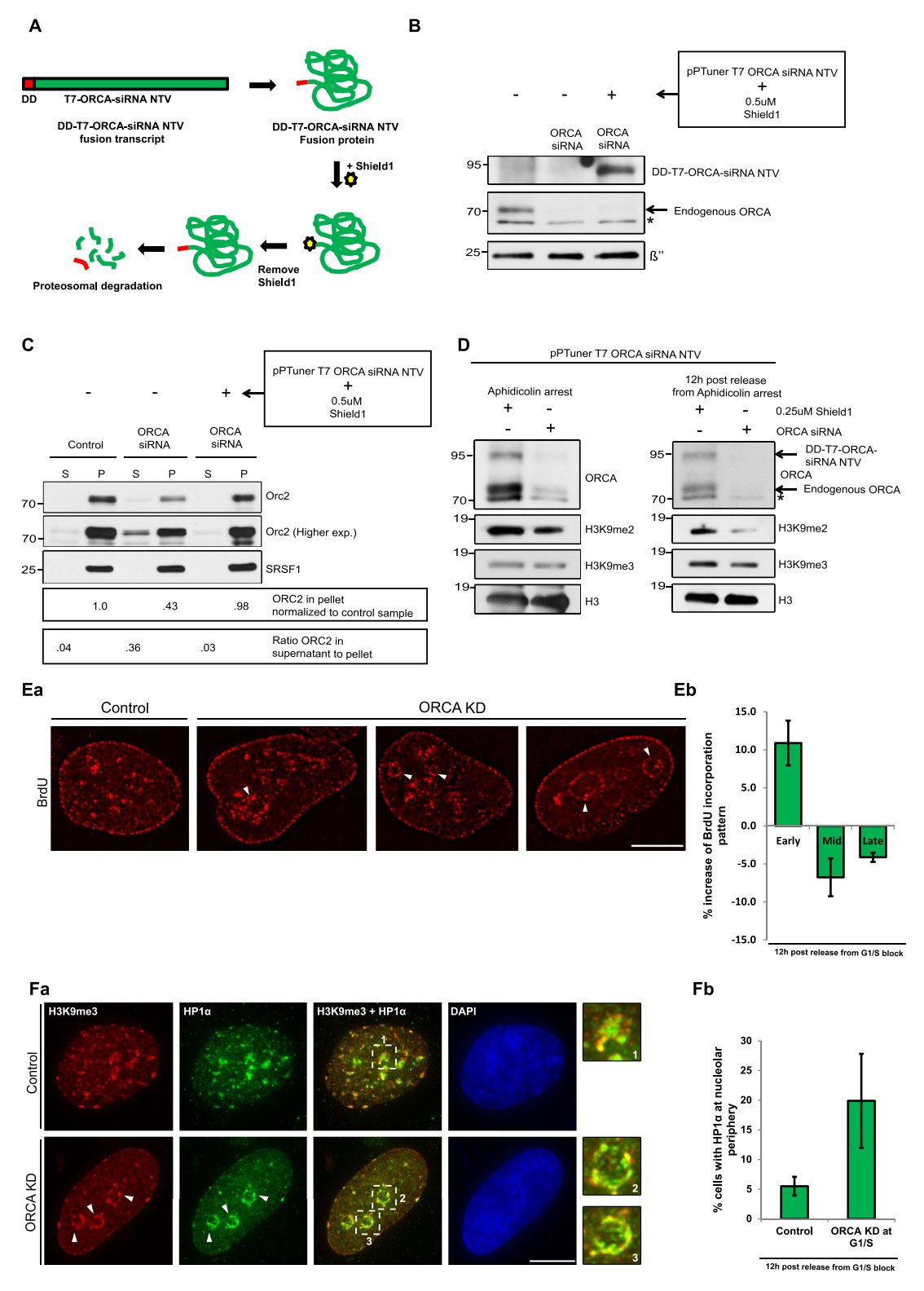

**Figure 8**. Heterochromatin organization role of ORCA is independent of its role in preRC assembly. (**A**) Schematic of depletion of ORCA using the proteotuner system. (**B**) Western blotting showing the levels of endogenous and exogenous ORCA in the presence of control and ORCA siRNA. β'', a nuclear speckle protein, serves as the loading control. Note that the DD-T7-ORCA-siRNA non-targetable version (NTV) is stabilized upon the addition of Shield1. (**C**) Chromatin fractionation and IB showing the levels of chromatin bound Orc2 in control and ORCA siRNA-treated cells (either in the absence or

*Figure 8. continued on next page*

*Figure 8. Continued*

presence of exogenous ORCA). Note the reduction in chromatin bound Orc2 in the absence of ORCA and the rescue of its levels upon expression of exogenous ORCA. Also note the increase in the soluble pool of Orc2 in the absence of ORCA and the decrease of its levels upon expression of exogenous ORCA. Splicing factor, SRSF1 is shown as a loading control. (**D**) IB showing the levels of endogenous and exogenous ORCA at G1/S and 12 hr post-release from aphidicolin. H3 is used as loading control. (**E**) **a–b** Patterns of BrdU incorporation in control and ORCA-depleted cells in late S-phase. The white arrowheads indicate preferential incorporation of BrdU incorporation at perinucleolar regions upon loss of ORCA. Scale bar, 10 μm. **b** % increase in S-phase cells displaying early and % decrease of the mid and late replication patterns in ORCA-depleted cells compared to control cells. Error bars represent s.d., n = 3 independent experiments with ~450 BrdU positive cells scored in each. (**F**) **a** H3K9me3 and HP1α immunofluorescence in control and ORCA-depleted cells. The white arrowheads indicate H3K9me3 and HP1α immunofluorescence at perinucleolar regions upon loss of ORCA. Representative regions in control and ORCA-depleted cells marked by white dotted squares (1, 2, and 3) are shown at 3× magnification on the right. Scale bar, 10 μm. **b** The % of cells with HP1α at nucleolar periphery in control and ORCA-depleted cells. Error bars represent s.d., n = 3.
The following figure supplement is available for figure 8:

**Figure supplement 1**. BrdU incorporation preferentially at perinucleolar regions in cells lacking ORCA.

In these cells, we examined the replication patterning by BrdU immunofluorescence. We observed a decrease in cells showing mid- and late- patterns of DNA replication and a concomitant increase in cells showing early patterns (*Figure 8Eb*), similar to our previous observations (*Figure 7B*). In cells lacking ORCA, a large number of cells showing mid-replication patterns showed BrdU staining preferentially at perinucleolar regions (*Figure 8Ea*, *Figure 8—figure supplement 1B*). Furthermore, we observed that H3K9me3 and HP1α were mislocalized and formed perinucleolar rings in cells lacking ORCA (*Figure 8Fa,Fb*, *Figure 8—figure supplement 1C*). Such localization was reminiscent of HP1α localization in Orc1- and Orc5-depleted cells (*Prasanth et al., 2010*). Moreover, both control and ORCA-depleted cells progressed through S-phase at comparable rates (*Figure 8—figure supplement 1D*) indicating that the observed defects in HP1α localization and BrdU incorporation upon loss of ORCA are due to defects in heterochromatin and not due to indirect effects of defects in S-phase progression. Based on the results, we propose that the observed defects in heterochromatin organization in ORCA-depleted post-G1 cells are independent of its known functions in preRC assembly.

## Discussion

ORCA, a key player in the initiation of DNA replication, associates with multiple components of the pre-replicative complex (*Shen et al., 2012*). The ORCA-ORC complex associates with heterochromatin, including telomeric and centromeric regions, even after replication has been accomplished suggesting that ORCA-ORC complex may play key roles in heterochromatin organization in addition to its role in pre-RC. The WD-repeat-containing domain (also found in ORCA) mediates interaction of proteins with nucleosomes/histones (*Suganuma et al., 2008*). For example, WDR5, a component of the MLL/SET1 KMT complex, binds to H3K4me2 using WD repeats (*Ruthenburg et al., 2006*). Similarly, HIRA, a WD-repeat-containing protein, binds to core histones and controls transcription (*Lorain et al., 1998*). Here, we demonstrate that ORCA associates with multiple H3K9 KMTs, binds to methylated H3K9, and regulates both the organization and replication of repressed chromatin marked with H3K9me2 and H3K9me3. Recently, a H3K9 KMTs multimeric complex has been identified that has been shown to be recruited to major satellite repeats and a subset of promoters of G9a-repressed genes and a functional cooperation of these enzymes is crucial for the regulation of G9a target genes (*Fritsch et al., 2010*). We demonstrate that ORCA-ORC associates with the H3K9 KMT-containing complex and in the absence of ORCA, this complex disintegrates. The loss of the enzymatic activity of this complex causes changes in the H3K9me2 and H3K9me3 profile in human cells. Based on this, we propose that ORCA is a scaffold protein that stabilizes the H3K9 KMT complex.

Recent work suggests that in mouse cells ORCA associates with pericentric heterochromatin via its association to H3K9me3 and maintains silencing at the major satellite repeats (*Chan and Zhang, 2012*). Based on our results, we speculate that the changes in transcription of satellite repeats upon ORCA-depletion are likely caused by the changes in the heterochromatin structure.

Immunoprecipitation experiments demonstrate that the WD-repeat of ORCA and the ankyrin repeat of G9a are crucial for the interaction between these two proteins. Ankyrin repeats of G9a also

contain methyl-lysine binding modules and can therefore generate as well as read the same epigenetic mark (*Collins et al., 2008*). We have observed that the SET domain of G9a and its catalytic activity is essential for the binding of ORCA to heterochromatin, suggesting that the chromatin modifications initiated by the KMT provide docking sites for ORCA. These in turn may facilitate the recruitment of accessory factors that stabilize the interaction and help the propagation of heterochromatin. We propose that ORCA recognizes repressive histone marks, binds to KMTs that in turn facilitate the propagation of the histone mark. The newly established marks then become docking sites for ORCA and the whole process is repeated and this results in the spreading of heterochromatin (*Figure 9A*).

Tri-methylation of H3K9, mono-methylation of H3-K27, and tri-methylation of H4-K20 are enriched at pericentric heterochromatin (*Peters et al., 2003*; *Rice et al., 2003*; *Schotta et al., 2004*). It is well established that H3K9 tri-methylation is a prerequisite for the subsequent H4K20 tri-methylation at the pericentric heterochromatin and this in turn sets the chromatin for binding of other key heterochromatin proteins including HP1 (*Lachner et al., 2001*; *Fischle et al., 2005*; *Stewart et al., 2005*). It is interesting to note that ORCA also interacts with Suv420H1 and H2, enzymes that catalyze H4K20 di- and tri-methylation, respectively (data not shown). It has been previously reported that Suv420H2 is a structural component of the heterochromatin and is required for chromatin compaction as well as cohesin recruitment (*Hahn et al., 2013*). Recently, Reinberg and co-workers have proposed

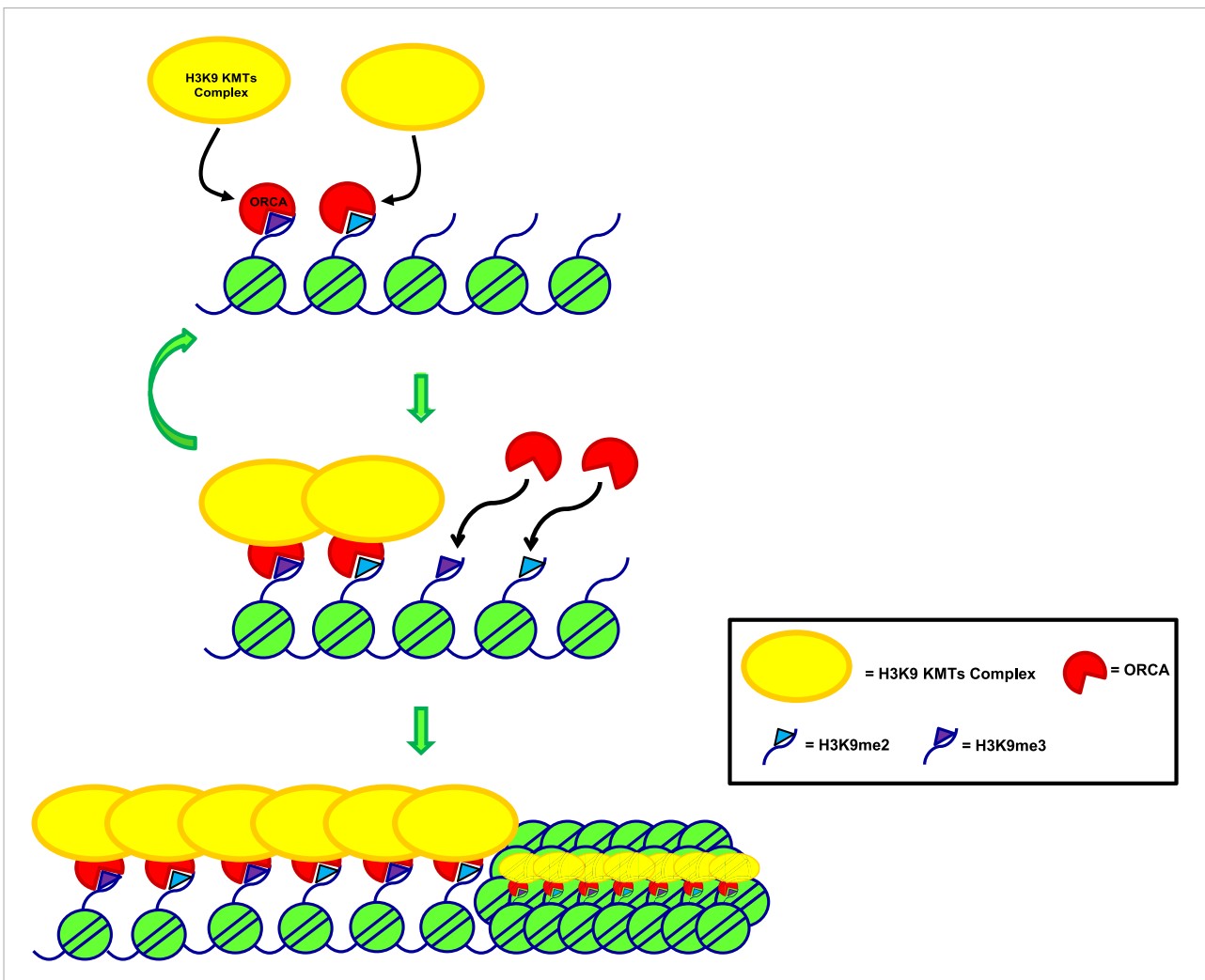

**Figure 9**. Model depicting the role of ORCA in organizing heterochromatin. Model representing mode of regulation of heterochromatin by ORCA.

that the H4K20 me1/2/3 is also crucial for the regulation and timing of replication origin firing and that ORCA and Orc1 directly recognize these chromatin sites (*Beck et al., 2012*). We are currently addressing the functional relevance of ORCA and Suv420H1/2 interaction in heterochromatin organization and replication progression.

Work from Jenuwein and co-workers have pointed to the idea that Prdm3 and Prdm16, H3K9 mono-methyltransferases, are also required for mammalian heterochromatin formation (*Pinheiro et al., 2012*). Similarly, mono-methylation of H3 at K9 catalyzed by SETDB1 has been shown to be a favored substrate for Suv39h for K9 tri-methylation, which can then establish heterochromatin (*Loyola et al., 2009*). The mechanism by which the Suv39h or H3K9me3 is targeted to pericentromere has been a long-standing question. It is generally assumed that HP1 is a key regulator of heterochromatin organization that is required for establishment and maintenance of this compacted form of chromatin. Spreading of the heterochromatin is thought to involve a mechanism where HP1 dimerizes, interacts with Suv39h1/2, and also recruits de novo DNA methyltransferase activity (*Almouzni and Probst, 2011*). The fact that HP1 associates with heterochromatin in a transient manner has suggested that other perhaps constitutively bound factors contribute to the organization of heterochromatin (*Cheutin et al., 2003*). In addition, recent work has demonstrated that pericentric heterochromatin can be generated independent of Suv39h-HP1 binding (*Muramatsu et al., 2013*). A transcription factor-based mechanism has also recently been suggested as an intrinsic mechanism for the formation of heterochromatin in mouse cells (*Bulut-Karslioglu et al., 2012*). Based on our results, we propose that in human cells, ORCA facilitates the recruitment, accumulation, and also the propagation of the heterochromatin by a self-sustaining loop mechanism, whereby ORCA binds to specific chromatin marks, associates with Suv39H1, that in turn propagates more H3K9me3 marks, generating more docking sites for ORCA (*Figure 9*).

We previously demonstrated that ORCA can facilitate the binding of ORC to chromatin and in the absence of ORCA, the binding of ORC to chromatin is compromised (*Shen et al., 2010*). However, it remains to be determined if the loss of chromatin-bound ORC in ORCA-depleted cells occurs at specific origins or at heterochromatic sites or both. Our data unequivocally demonstrate that ORCA binds to H3K9-methylated chromatin and facilitates the recruitment of KMTs as well as other components of ORC and MCMs to these sites. It is possible that the regulation of these epigenetic modifications by ORCA may provide identity to repressed chromatin and this in turn is crucial for proper replication. This idea is supported by our observation that ORCA associates with repressive KMTs only in a heterochromatic environment. It is known that G9a mediates di-methylation of H3K9 at late-replicating chromatin and this occurs predominantly at the nuclear periphery. H3K9me2 and H3K9me3 are enriched at the late-replicating facultative or constitutive heterochromatin, respectively (*Wu et al., 2005*). The reduction of these marks upon loss of ORCA leads to changes in the replication timing only of these regions as indicated by the significant decrease in late-replication patterns upon ORCA KD. This is very similar to previous reports that show that loss of Rif1 causes reduction in mid-replication patterns and global changes in replication timing primarily due to Rif1's role in organizing chromatin loops (*Cornacchia et al., 2012*; *Yamazaki et al., 2012*; *Kumar and Cheok, 2014*). ORCA could be functioning in a similar fashion as an organizer of heterochromatin, and therefore, in cells lacking ORCA replication timing is altered. Furthermore, using the proteoTuner system, we demonstrate that the role of ORCA in chromatin organization is independent of its role in preRC assembly.

Heterochromatin is formed as a result of a maturation process that requires several steps and ORCA acts early in this process. Our results demonstrate that ORCA is a chromatin reader that facilitates the assembly of the writer KMT complex and its associated partners to specific chromatin sites. These together regulate key cellular events, including DNA replication and heterochromatin organization.

## Materials and methods

### Plasmid constructs

Human G9a (hG9a) full-length and mutant clones were obtained using PCR using pCDNA3 Flag G9a plasmid provided very kindly by Dr Martin Walsh. The mutants were cloned into pCGN, pEYFP, pECFP and pEmCherry vectors (Clonetech, Mountain View, CA), and pEYFP-LacI vector. pEGFP G9a full-length and ΔSET constructs were also kindly provided by Dr Walsh. Mouse G9a (mG9a) full-length was amplified from pSV2 YFP mG9a (Dr David Spector's lab) (*Janicki et al., 2004*). pEYFP LacI mG9a

ΔSET and H1166K constructs were cloned by amplification from respective pSG5 HA mG9a constructs kindly provided by Dr Michael Stallcup. Flag GLP was kindly provided by Dr Dan Levy.

Human Suv39H1 full-length and mutant clones were obtained using PCR using Flag-Suv39H1 plasmid provided very kindly by Dr Rama Natarajan (*Villeneuve et al., 2008*). The mutants were cloned into pCGN vector and pEYFP vector (Clonetech, Mountain View, CA). pSV2-YFP-mSuv39H1 has been described previously (*Janicki et al., 2004*).

Myc-EZH2 was kindly provided by Dr Francois Fuks, Free University Brussels. EZH2 was PCR amplified and cloned into pEYFP-LacI vector.

Flag Suv420H1.1 and H2 constructs were kindly provided by Dr Craig Mizzen.

pEGFP-LacI vector was a kind gift from Dr Miroslav Dundr (*Kaiser et al., 2008*) and used for making pEYFP-LacI vector. T7-ORCA mutants, pEYFP and CFP ORCA, pECFP-LacI and pECFP-LacI-ORCA have been described previously (*Shen et al., 2010*). YFP-LacI-Orc2 was cloned by amplifying and inserting Orc2 into pEYFP-LacI vector. YFP-Orc1 construct has been described previously (*Hemerly et al., 2009*).

T7-ORCA-siRNA NTV was cloned into pPTuner IRES2 (Clonetech, Mountain View, CA) vector.

All the cloned constructs were confirmed by sequencing and used for immunoprecipitation and/or cell biological experiments.

## Cell culture

U2OS cells were grown in Dulbecco's modified Eagle's medium (DMEM) containing high glucose and supplemented with 10% fetal bovine serum (FBS—HyClone GE, Pittsburg, PA). WI38 cells were also grown in the same medium but supplemented with non-essential amino acids. Hela suspension cells (Hela-XLP GLP) were grown in DMEM supplemented with 5% fetal calf serum and penicillin-streptomycin. U2OS-2-6-3 CLTon cells were grown in DMEM with 10% Tet system approved FBS (Clonetech, Mountain View, CA).

For arresting cells at G1/S, aphidicolin (stock 10 mg/ml) was added to the cells at a final concentration of 5 μg/ml for 12 hr. Cells were then washed three times with PBS and released into S-phase with medium (DMEM + 10%FBS) lacking aphidicolin. Cells were then collected at 4, 8, and 12 hr post-G1/S block for immunofluorescence and flow cytometry analysis.

siRNA transfection for ORCA depletion: cells were grown to 30% confluency and siRNA against ORCA or control luciferease gene (*Shen et al., 2010*) was delivered into the cells at a final concentration of 100 nM using Lipofectamine RNAimax (Invitrogen, Carlsbad, CA). The siRNAs were delivered twice at the gap of 24 hr, and the cells were collected 24 hr after the second round of transfection for subsequent analysis.

## Rescuing of ORC loading by using DD-T7-ORCA-siRNA NTV

U2OS cells were transfected with of DD-T7-ORCA-siRNA NTV construct along with siRNA against ORCA. 5 hr later Shield1 (0.5 μM) was added to the medium. 24 hr after the first round of KD, a second round of ORCA siRNA treatment was carried out in the presence of Shield1. Samples were collected 24 hr later for chromatin fractionation to examine ORC loading.

## Depletion of ORCA using proteotuner

As described above, DD-T7-ORCA-si NTV (500 ng plasmid was transfected) was expressed in U2OS cells grown on coverslips in the presence of Shield1 (0.25 μM). This was followed by the addition of fresh medium containing aphidicolin (5 μg/ml) + Shield1 (0.25 μM). 14 hr post-aphidicolin block, cells were washed thrice with PBS containing aphidicolin, with or without Shield1, respectively. This was followed by performing control and ORCA depletions in Opti-MEM containing aphidicolin, with or without Shield1. The control and ORCA-depleted cells were washed with PBS containing or lacking Shield1, respectively. Then, the cells were released into S-phase using medium with or without Shield1 for control and ORCA-depleted cells followed by late S sample collection 12 hr later for Western blotting and immunofluorescence analysis.

## Insect cell culture and baculovirus expression

His ORCA, G9a, and Suv39H1 viruses were generated by using pFastBac HT-B-His-tagged-ORCA, G9a, and Suv39H1, respectively (*Shen et al., 2012*) (by following Bac-to-Bac baculovirus expression system—Invitrogen). Virus production was carried out in *Sf9* cells with viruses collected 72 hr

post-infection (multiplicity of infection 5 to 10). Protein expression was carried out in *Hi5* cells by collecting cells 65 hr post-infection. Nuclei were collected by using Hypotonic lysis buffer (20 mM potassium phosphate buffer pH 7.5, 5 mM KCl, 1.5 mM $MgCl_2$, and 5 mM b-mercapoethanol) making nuclear extracts in PK50 buffer (20 mM potassium phosphate buffer pH 7.5, 50 mM KCl, 0.02% NP-40, 10% glycerol, 5 mM b-mercaptoethanol with protease and phosphatase inhibitors) (*Siddiqui and Stillman, 2007*). 45% ammonium sulfate precipitation was carried out followed by reconstitution of His-ORCA, G9a, and Suv39H1 in PK50 buffer.

## Immunofluorescence

Cells were fixed with 2% formaldehyde in phosphate buffered saline (PBS, pH 7.4) for 15 min in room temperature (RT) followed by permeabilization with 0.5% Triton X-100 in PBS for 7 min on ice or pre-extracted before fixing with 0.5% Triton X-100 in Cytoskeletal buffer (CSK: 100 mM NaCl, 300 mM Sucrose, 3 mM $MgCl_2$, 10 mM PIPES at pH 6.8) for 5 min on ice. Blocking was then done for 30 min with 1% Normal goat serum (NGS) in PBS. Primary antibody incubation was then carried out for 1 hr in a humidified chamber followed by secondary antibody incubation for 25 min. The cells were then stained with DAPI (4′,6-Diamidino-2-Phenylindole) and mounted using VECTASHIELD (Vector Laboratories Inc., Burlingame, CA). The following antibodies were used for immunofluorescence: BrdU (1:500; mAb BU-33, Sigma, St. Louis, MO), Lamin (1:750), H3K9me2 (1:100; 07-212, Millipore, Billerica, MA), H3K9me3 (1:200, Millipore 07-523), HP1α (1:100, Millipore 3584).

BrdU immunofluorescence: after primary and secondary antibody incubation for lamin immuno-fluorescence, cells were fixed with 2% formaldehyde solution in PBS. This was followed by acid denaturation of DNA using 4 N HCl for 25 min at RT. Three washes with PBS and two washes with PBS-NGS followed. This was followed by incubation of the cells with anti-BrdU antibody followed by secondary antibody incubation and mounting.

Zeiss Axioimager z1 fluorescence microscope (Carl Zeiss Inc., Jena, Germany) equipped with chroma filters (Chroma technology, Bellows Falls, CA) was used for observing the cells and statistics. Axiovision software (Zeiss) was used for digital imaging using Hamamatsu ORCA cooled CCD camera. Cells were also examined on the Delta vision optical sectioning deconvolution instrument (Applied precision, Pittsburgh, PA) on an Olympus microscope.

## Immunoprecipitations and immunoblots

For co-IP with transiently transfected HKMTs and ORCA, co-transfections were carried out in U2OS cells. Cells were lysed, 24 hr post-transfection, in IP buffer (50 mM Tris pH 7.4, 150 mM NaCl, 10% glycerol, 0.1% NP-40, 1 mM DTT (Dithiothreitol) supplemented with the protease and phosphatase inhibitors). After pre-clearing with Gammabind Sepharose beads for 1 hr, the lysates were incubated with appropriate antibody-conjugated agarose overnight. The beads were washed in the same IP buffer and finally denatured by the addition of Laemmli buffer. The complex was analyzed by Western blotting.

For immunoprecipitations and IB the following antibodies were used anti-GFP (1:500; Covance, Princeton, NJ), anti-Flag M2 (1:500, Sigma), anti-HA 12CA5 (1:100) and anti-T7 (1:5000; Novagen, Billerica, MA), anti-ORCA pAb 2854-1 AP (1:500), anti-G9a (1:500, Sigma), anti-Suv39h1 (1:200, Cell Signaling, Danvers, MA), anti-ORC2 pAb 205-6 (1:1000), anti-Geminin (1:1000, Santa Cruz, Dallas, TX), anti-MCM3 TB3 (1:750), anti-α-tubulin (1:10,000, Sigma–Aldrich), anti-H3K9me2 (1:200, Millipore 07-212), anti-H3K9me3 (1:500, Millipore 07-523), anti-SF2 AK96 (1:750), anti-PCNA mAb PC10 (1:750) antibodies.

For IP in the presence of EtBr, lysates were made with IP buffer described above followed by addition of EtBr (stock 10 mg/ml, working 50 µg/ml). After pulldown, beads were washed with IP buffer supplemented with 80 µg/ml of EtBr. For Benzonase treatment, cells (grown in 6-cm plates) were lysed for 10 min in IP buffer (50 mM HEPES pH 7.9, 10% glycerol, 200 mM NaCl, 0.1% Triton X-100,1 mM $MgCl_2$) supplemented with protease and phosphatase inhibitors. 1000 U of Benzonase (Sigma) was then added followed by nutation at RT for 20 min. EDTA (final concentration 5 mM) was added to stop the reaction followed by centrifugation at 12,500 rpm, 5 min at 4°C. The supernatant was used for subsequent antibody incubation.

## Nuclear extracts for semi-endogenous IP

The nuclear extraction protocol has been described previously (*Robin et al., 2007*; *Fritsch et al., 2010*). We carried out HA immunoprecipitation in HeLa cells stably expressing Flag-HA-GLP and

Flag-HA-G9a stable by retroviral transduction. First, nuclear extract was made using an equivalent of 20 g of dry cell pellet, which approximately corresponds to 3 billion cells. Cells were resuspended in hypotonic buffer (10 mM Tris pH 7.6, 1.5 mM $MgCl_2$, 10 mM KCl) with the volume of hypotonic buffer being equal to the packed volume of cells. The suspension was then dounced 20 times using tight pestle followed by adding one third the packed volume of sucrose buffer (20 mM Tris pH 7.6, 15 mM KCl, 60 mM NaCl, 0.34 M Sucrose, 0.15 mM Spermine, 0.5 mM Spermidine). Then, centrifugation was carried out (9000 rpm, 7 min). The supernatant was discarded, and the nuclei were resuspended in low salt buffer (20 mM Tris pH 7.6, 25% glycerol, 1.5 mM $MgCl_2$, 0.2 mM EDTA, 20 mM NaCl). This was followed by adding high salt buffer (20 mM Tris pH 7.6, 25% glycerol, 1.5 mM $MgCl_2$, 0.2 mM EDTA, 900 mM NaCl) dropwise while vortexing to make the final salt concentration 300 mM. After incubation on ice for 30 min, one third the volume of sucrose buffer was added followed by centrifugation at 1000 rpm, 10 min, 4°C. The supernatant is the nuclear soluble fraction. The pellet (chromatin bound fraction) was resuspended in sucrose buffer and $CaCl_2$ (final concentration 1 mM) was added. The sample was then preheated for 1 min at 37°C and MNase was added to a final concentration of 0.0025 U/μl. The sample was then incubated for exactly 12 min at 37°C followed by placing the tubes on ice. 0.5 M EDTA was added at a final concentration of approximately 3.6 μM. The samples were then sonicated (Bioruptor, high amplitude 5cycles: each cycle 1 min ON, 1 min OFF). Protease and phosphatase inhibitors were added, and the samples (nuclear soluble and chromatin bound fractions) were ultracentifuged at 40,000 rpm for 30 min. The supernatants were transferred to a fresh tube. Tagged-H3K9 HMTs complexes were then purified by immunoprecipitation using anti-FLAG antibody immobilized on agarose beads (cat# A2220, Sigma). After elution with the FLAG peptide (Ansynth, The Netherlands), the bound complexes containing nucleosomes were further affinity purified on anti-HA antibody-conjugated agarose (cat# A2095, Sigma) and eluted with the HA peptide (Ansynth, The Netherlands). The samples were then analyzed by IB.

## Nuclear extracts for single-molecule pulldown

Cells (HeLa for gel filtration and U2OS for SiMPull) were lysed in hypotonic buffer (10 mM HEPES-NaOH pH 7.9, 10 mM KCl, 2 mM $MgCl_2$, 0.34 M Sucrose, 10% glycerol, 0.1% Triton X-100) supplemented with 1 mM DTT, protease and phosphatase inhibitors. The lysate was incubated at 4°C for 5 min after which nuclei were collected by centrifuging at 1500×$g$ for 5 min. The pellet was then resuspended in nuclear extraction buffer (10 mM HEPES-NaOH pH 7.9, 2 mM $MgCl_2$, 1 mM EGTA, 25% glycerol, 350 mM NaCl, 0.1% Triton X-100, and 1 mM DTT) supplemented with protease and phosphatase inhibitors. The supernatant was collected by centrifugation at 12,000 rpm for 5 min. The lysate was then used for co-localization studies by SiMPull.

## Direct interaction studies

Baculovirally purified His-ORCA and His-G9s/His-Suv39H1 were diluted using PK 50 buffer and incubated together for 2 hr at 4°C in the presence of ORCA antibody or pre-bleed. ORCA containing complexes were then pulled down followed by washes with PK 150 buffer (20 mM potassium phosphate buffer pH 7.5, 150 mM KCl, 0.5% NP-40, 10% glycerol, 5 mM b-mercaptoethanol with protease and phosphatase inhibitors). The samples were finally denatured by the addition of Laemmli buffer. The complexes were analyzed by Western blotting.

## Flow cytometry—BrdU-PI staining

U2OS cells were grown in 6-cm plates to approximately 50% confluency followed by incorporation of BrdU (stock 10 mM; working 50 μM ) for 1 hr at 37°C. Cells were then harvested at 3500 rpm, 15 min followed by washing with 1% BSA (Bovine serum albumin) in PBS (Phosphate buffered saline) (pH 7.4). The cells were then resuspended in 0.9% NaCl (final cell density: $2 \times 10^6$ cells/ml). The cells were then fixed by adding chilled 100% ethanol to a final concentration of 50% (left overnight at −20°C). After spinning down the fixed cells, DNA was denatured by resuspending in 2 N HCl + 0.5% Triton X—100 and incubating for 30 min at RT. The cells were then pelleted and resuspended in 0.1 M Sodium tetraborate pH 8.5. This was followed by centrifugation at 3500 rpm, 15 min at 4°C followed by resuspending the cells in PBS + 1% BSA + 0.5% tween 20. Anti-BrdU FITC antibody (1ug Ab/$10^6$ cells) was added and the cells were incubated at RT for 1 hr. 1 ml of PBS + 1% BSA + 0.5% Tween 20 was added after that followed by spinning down the cells and proceeding with RNase A treatment and PI staining as described in the previous section.

## Peptide pulldown

Human Histone H3 (amino acids 1–15) peptides were synthesized (Biomer technology, Pleasanton, CA) with a Cysteine at the N terminus. The K9 (Lysine at position 9) of the peptides was unmodified, acetylated, mono-, di-, or tri-methylated. The peptides were dissolved in water, quantitated using reverse phase chromatography, lyophilized and stored at −20°C as 1-mg aliquots.

For coupling the peptides to SulphoLink Coupling Resin (Thermo Scientific, Waltham, MA), the peptides were reduced first. For this, 1 mg of each peptide was dissolved in 3 ml of coupling buffer (50 mM Tris pH 8.7, 5 mM EDTA, final pH adjusted to 8.5) supplemented with TCEP HCl (Thermo Scientific) and allowed to incubate at RT for 1 hr. 2 ml of the beads was washed with 5 ml coupling buffer (three 10 min washes) and resuspended in 1 ml of coupling buffer. 3 ml of the reduced peptides was added to the slurry followed by mixing immediately to distribute the peptide throughout the slurry. The mixture was incubated overnight at RT with gentle mixing. The beads were then spun down (2000 rpm, 2 min), washed with 6 ml of coupling buffer (three 5 min washes), resuspended in 5 ml coupling buffer + 1 ml L-Cysteine HCl. The mixture was incubated overnight at RT. The beads were then spun down and washed with 5 ml of 1 M NaCl (three 5 min washes). This was followed by washing two 5 min washes with 5 ml of storage solution (0.05% $NaN_3$ in water) and final suspension in 5 ml of storage solution and storage at 4°C.

50 µl packed volume of beads (300 µl of bead slurry) was washed with PK 150 buffer (20 mM potassium phosphate buffer pH 7.5, 150 mM KCl, 0.02% NP-40, 10% glycerol, 5 mM β-mercaptoethanol) and incubated with baculovirally purified His-ORCA for 2 hr at RT. This was followed with five washes with PK 150 buffer. The beads were then resuspended in Laemmli buffer and analyzed by Western blotting.

## Single molecule pulldown

SiMPull experiments were carried out in flow chambers prepared on quartz microscope slides, which were passivated with methoxy-polyethylene glycol (mPEG) doped with 1% biotin-PEG (Lysan Bio, Inc, Arab, Al) (Jain et al., 2011). Appropriate biotinylated antibody was immobilized on PEG passivated surfaces at approximately 20 nM concentration for 20 min after coating the flow chambers with 0.2 mg/ml NeutrAvidin for 5 min. Antibodies were immobilized on NeutrAvidin (Thermo)-coated flow chambers either by incubating with biotinylated T7 antibody (Novagen) for 10 min. RIPA buffer lysed samples were then incubated in the chamber for 20 min and washed twice with the buffer (10 mM Tris-HCl pH 8.0, 50 mM NaCl 0.1 mg/ml BSA). Single-molecule data were acquired by a prism-type TIRF microscope and analyzed using scripts written in Matlab. For ORCA-Orc1; ORCA-G9a; ORCA-Suv39H1 SiMPull analysis, lysates were made from the cells transiently transfected with T7-ORCA with YFP-Orc1, YFP-G9a, or YFP-Suv39H1, respectively. For multimeric complex assembly analysis using SiMPull, cells were transfected with T7-ORCA, YFP-Orc1, and mCherry-G9a or T7-ORCA, YFP-Suv39H1, and mCherry-G9a.

For peptide pulldown experiments, biotinylated peptides were immobilized instead of antibodies. Cells lysed in RIPA buffer or nuclear extracts (depending on the experiment) were then incubated in the flow chamber for 20 min followed by wash with T300 buffer (20 mM Tris-HCl, pH 8.0, 300 mM NaCl, 0.1 mg/ml bovine serum albumin [BSA]). Single molecules were visualized by prism-type TIRF microscope and analyzed using Matlab scripts (https://github.com/vasuagg/SiMPull_Analysis). Cell lysate was appropriately diluted in T300 buffer to obtain optimal single molecule density on the surface.

## SiMPull data analysis

Single molecule data were acquired as the average number of YFP or mCherry fluorescent molecules per imaging area (5000 µm²) as shown in the histograms. The error bars represent standard deviation of the mean values from 20 imaging areas. Number of fluorescence photobleaching steps was determined for each YFP-tagged molecule and accumulated to obtain the stoichiometry of the complex. Colocalization percentage between YFP and mCherry was calculated as the number of coaligned molecules of one fluorescent molecule with respect to the fluorescent molecules found in lower density on the surface. This was needed since the number of YFP and mCherry tagged proteins was not pulled down to the same extent due to their independent interaction with ORCA. Colocalization criterion was set at 2 pixels, which correspond to a diffraction limited spot (~300 nm)

for our TIRF setup. Error bars represent standard deviation of the mean values obtained from three independent experiments. For pulled down experiments performed using H3K9 peptides, the expression level of YFP-WT ORCA and YFP-mutant-ORCA was compared in the beginning by performing a direct pulldown by anti-GFP. The peptides pulldown was then performed at appropriate lysate dilution such that protein expression was same for WT (Wild type) and mutant ORCA.

## Chromatin immunoprecipitation

H3K9me2 and me3 ChIPs: formaldehyde (Sigma) was added to culture medium to a final concentration of 1%. Crosslinking was allowed to proceed for 10 min at RT and stopped by the addition of glycine at a final concentration of 0.125 M. Fixed cells were washed and harvested with PBS. Chromatin was prepared by two subsequent extraction steps (10 min at 4°C) with Buffer 1 (50 mM Hepes/KOH pH 7.5; 140 mM NaCl; 1 mM EDTA; 10% glycerol; 0.5% NP-40; 0.25% Triton) and Buffer 2 (200 mM NaCl; 1 mM EDTA; 0.5 mM EGTA; 10 mM Tris pH 8). Nuclei were then pelleted by centrifugation, resuspended in Buffer 3 (50 mM Tris pH 8; 0.1% SDS; 1% NP-40; 0.1% Na-Deoxycholate; 10 mM EDTA; 150 mM NaCl) and subjected to sonication with Bioruptor Power-up (Diagenode, Denville, NJ) yielding genomic DNA fragments with a bulk size of 150–300 bp. Chromatin was precleared with Protein A/G ultralink beads (53,133, Pierce, Grand Island, NY) for 2 hr at 4°C and immunoprecipitation with the specific antibodies carried out overnight at 4°C. Immune complexes were recovered by adding pre-blocked protein A/G ultralink beads and incubated for 2 hr at RT. Beads were washed twice with low salt buffer (0.1% SDS; 1% Triton; 2 mM EDTA; 20 mM Tris pH 8; 150 mM NaCl), twice with high salt buffer (0.1% SDS; 1% Triton; 2 mM EDTA; 20 mM Tris pH 8; 500 mM NaCl), once with LiCl wash buffer (10 mM Tris pH 8.0; 1% Na-deoxycholate; 1% NP-40, 250 mM LiCl; 1 mM EDTA), and twice with TE + 50 mM NaCl. Beads were eluted in TE + 1% SDS at 65°C, and cross-link was reversed O/N at 65°C. The eluted material was phenol/chloroform extracted and ethanol precipitated. DNA was resuspended in water and q-PCR performed using PowerSYBR Green PCR Master mix (Applied Biosystems, Pittsburgh, PA) and analyzed on a 7300 PCR System (Applied Biosystems). ChIP-qPCR results were represented as percentage (%) of IP/input signal (% input). HA-ORCA ChIPs were carried out using HA-ORCA stable cell lines in U2OS using a similar protocol with the following modifications. All the washing steps after immune complexes pulldown were done once followed by two washes with TE. Beads were eluted with 1% SDS + 0.1 M NaHCO3 at 65°C, and cross-link was reversed O/N at 65°C. The eluted material was purified using Qiagen gel purification kit and qPCR carried out.

G9a and Suv39H1 ChIPs were carried out using double-crosslinking protocols. The first crosslinking was carried out using disuccinimidyl glutarate (DSG) (Santa Cruz; stock 50 mM DSG in DMSO) and the second crosslinking using formaldehyde. U2OS cells were grown in 10-cm plates to 80% confluency, washed twice with PBS (pH 7.4). Freshly made crosslinking solution (2 mM DSG + 1 mM MgCl$_2$ in PBS-pH 8.0) was added for 45 min at RT. The cells were then washed twice with PBS (pH 7.4) and 10 ml of freshly made crosslinking solution (1% formaldehyde, 15 mM NaCl, 150 μM EDTA, 75 μM EGTA, 15 μM HEPES pH 7.9) was added for 10 min at RT. Then 3 ml of freshly made 1 M Glycine was added for 5 min at RT followed by two cold washes with PBS (pH 7.4). The cells were then pelleted in PBS (supplemented with protease inhibitors) followed by lysis with 300 μl of SDS lysis buffer (1% SDS, 10 mM EDTA, 50 mM Tris pH 8.0). The lysate was then subjected to sonication with Bioruptor Power-up (Diagenode). Chromatin was precleared with Dynabeads protein G (Life Technologies, Grand Island, NY) for 2 hr at 4°C and immunoprecipitation with the specific antibodies carried out overnight at 4°C. Immune complexes were recovered by adding pre-blocked Dynabeads (1 mg/ml BSA, 0.4 mg/ml salmon sperm DNA) and incubated for 2 hr at 4°C. Beads were washed once with low salt buffer (0.1% SDS; 1% Triton; 2 mM EDTA; 20 mM Tris pH 8; 150 mM NaCl), once with high salt buffer (0.1% SDS; 1% Triton; 2 mM EDTA; 20 mM Tris pH 8; 500 mM NaCl), once with LiCl wash buffer (10 mM Tris pH 8.0; 1% Na-deoxycholate; 1% NP-40, 250 mM LiCl; 1 mM EDTA), and twice with TE (10 mM Tris pH 8.0, 1 mM EDTA). Beads were eluted in elution buffer (1% SDS, 0.1 M sodium bicarbonate in water) at 65°C twice, 10 min each. The eluates were pooled (250 μl), NaCl added (final concentration 0.2 M) and cross-link was reversed O/N at 65°C. The eluted material was Rnase A treated (10 μg/ml, 1 hr at 37°C) followed by proteinase K treatment (4 μl 0.5 M EDTA, 8 μl 1 M Tris pH 6.9, 1 μl proteinase K 20 mg/ml) at 42°C for 2 hr. DNA was purified using QIAquick PCR purification kit (Qiagen) and q-PCR performed SYBR Green PCR Master mix and analyzed on a 7300

PCR System (Applied Biosystems). ChIP-qPCR results were represented as percentage (%) of IP/input signal (% input).

## BrdU ChIP after ORCA KD and data analysis

Two rounds of ORCA KD were carried out, 24 hr apart. The cells were then arrested using aphidicolin for 12 hr followed by release into S-phase and samples were collected 0, 4, 8, and 12 hr post release for BrdU ChIP. Prior to each time point collection, cells were pulsed for 2 hr with BrdU (10 µM).

Cells for each time point were then lysed with 300 µl of SDS lysis buffer (1% SDS, 10 mM EDTA, 50 mM Tris pH 8.0). The lysates were subjected to sonication with Bioruptor Power-up (Diagenode). 100 µl sheared chromatin aliquots were then placed on 95°C heat block for 10 min. This was followed by snap chilling the samples for 10 min. The samples were then diluted, precleared and processed further in a manner identical to the ChIP protocol described in the previous section.

qPCRs were carried out with purified DNA of input, BrdU ChIP, and mouse IgG ChIP samples obtained at 0, 4, 8, and 12 hr post-aphidicolin release time points. The qPCR signals of BrdU and mouse IgG samples were calculated as percent input values. Fold enrichment of BrdU ChIP over mouse IgG ChIP was calculated for each time point.

## Construction and sequencing of ChIPSeq libraries

H3K9me2 and me3 ChIP were performed as above. Five to fifteen nanograms of ChIP DNA or un-enriched whole cell extract (Input) were prepared for sequencing on an Illumina HiSeq2000. Libraries were constructed with the Truseq DNA sample prep kit V2 (Illumina, San Diego, CA) with the following modifications: 10 ng of ChIP DNA were used as input material. DNA fragments were blunt-ended, 3′-end A-tailed and ligated to indexed TruSeq adaptors. The adaptors were diluted in a ratio of 1:20 to adjust for the input amount of DNA. Indexed adaptors allow for sequencing of multiple samples on the same lane (multiplexing). The adaptor-ligated ChIP DNAs were individually size selected on a 2% agarose gel (Ex-Gel, Life Technologies, CA) to obtain the ligated fragments 300–800 bp in length. Size-selected DNAs were amplified by PCR to selectively enrich for those fragments that have adapters on both ends. Amplification was carried out for 15 cycles with the Kapa HiFi polymerase (Kapa Biosystems, Woburn, MA) to reduce the likeliness of multiple identical reads due to preferential amplification. The final libraries were quantitated by qPCR on an ABI 7900, to allow for accurate quantitation and maximization of number of clusters in the flowcell. Final amplified libraries were also run on Agilent bioanalyzer DNA 7500 LabChips (Agilent, Santa Clara, CA) to determine the average fragment size and to confirm the presence of DNA of the expected size range.

The libraries were pooled and loaded onto a lane of an 8-lane flowcell for cluster formation and sequenced on an Illumina HiSeq2000. The libraries were sequenced from one end of the molecules to a total read length of 100 nt. The raw .bcl files were converted into demultiplexed compressed. fastq files using CASAVA 1.8.2.

The complete ChIP-seq data are available at http://www.ncbi.nlm.nih.gov/geo/query/acc.cgi?acc=GSE68129.

## Acknowledgements

We thank members of the Prasanth laboratory for discussions and suggestions. We thank Drs M Stallcup, D Levy, R Natarajan, F Fuks, M Dundr, A Hernandez, R Donthu, C Fields, D Spector, and B Stillman for providing reagents and suggestions. We would like to thank Drs D Forsthoefel, S Ceman and D Rivier for critical reading of the paper. This work was supported by NSF-CMMB-IGERT and F31 (CA180616) NIH fellowship to SG; ACS (RSG11-174-01RMC) and NIH (1RO1GM088252) award to KVP and NSF career (1243372) and NIH (1RO1GM099669) awards to SGP. TH is an investigator with Howard Hughes Medical Institute. Work in SA group was supported by the: Agence Nationale de la Recherche (H3K9-methylome and EPILIS grants), Association Française contre les Myopathies (AFM); Fondation ARC; GEFLUC; Institut National du Cancer (INCa, grant 2012-1-PLBIO). JP was recipient of PhD fellowships from the MESR and Fondation ARC. The authors declare no competing financial interests.

## Additional information

### Competing interests

TH: Reviewing editor, *eLife*. The other authors declare that no competing interests exist.

## Funding

| Funder | Grant reference | Author |
| --- | --- | --- |
| National Institutes of Health (NIH) | F31 (CA180616) fellowship | Sumanprava Giri |
| National Institutes of Health (NIH) | 1RO1GM088252 | Kannanganattu V Prasanth |
| National Science Foundation (NSF) | 1243372 | Supriya G Prasanth |
| National Institutes of Health (NIH) | 1RO1GM099669 | Supriya G Prasanth |
| Howard Hughes Medical Institute (HHMI) | Investigator | Taekjip Ha |
| Agence Nationale de la Recherche | H3K9-methylome and EPILIS grants | Slimane Ait-Si-Ali |
| AFM-Téléthon (French Muscular Dystrophy Association) | | Slimane Ait-Si-Ali |
| Fondation ARC pour la Recherche sur le Cancer | | Slimane Ait-Si-Ali |
| Institut national du cancer | (INCa, grant 2012-1-PLBIO) | Slimane Ait-Si-Ali |
| Ministère de l'Education Nationale, de l'Enseignement Superieur et de la Recherche | PhD fellowships | Julien Pontis |
| Fondation ARC pour la Recherche sur le Cancer | PhD fellowships | Julien Pontis |
| National Science Foundation (NSF) | CMMB-IGERT | Sumanprava Giri |
| American Cancer Society | RSG 11-174-01RMC | Kannanganattu V Prasanth |

The funders had no role in study design, data collection and interpretation, or the decision to submit the work for publication.

## Author contributions

SG, VA, TH, Conception and design, Acquisition of data, Analysis and interpretation of data, Drafting or revising the article; JP, SA-S-A, Acquisition of data, Analysis and interpretation of data, Drafting or revising the article; ZS, AC, Conception and design, Acquisition of data, Analysis and interpretation of data; AK, Conception and design, Analysis and interpretation of data, Contributed unpublished essential data or reagents; CM, Acquisition of data, Analysis and interpretation of data; KVP, Conception and design, Analysis and interpretation of data, Drafting or revising the article; SGP, Conception and design, Acquisition of data, Analysis and interpretation of data, Drafting or revising the article, Contributed unpublished essential data or reagents

# Additional files

## Supplementary files

• Supplementary file 1. H3K9me3 ChIP-seq peaks in control and ORCA-depleted samples.

• Supplementary file 2. Primers used for validation of ChIP-seq. Our attempts on H3K9me2 ChIP-seq did not succeed because of the technical challenge associated with sequencing the broad H3K9me2 peaks. Similar problems with H3K9me2 ChIP-seq have been previously reported by other studies (*Yuan et al., 2009*). As an alternate, regions that showed significant reduction of H3K9me3 in the ChIP-seq experiment (as evident by the wiggle plots; *Figure 5Da–Dd* and *Figure 5—figure supplement 1Ba,Bb*) were chosen for H3K9me2 ChIP-qPCR validation (*Supplementary file 2*).

## Major dataset

The following dataset was generated:

| Author(s) | Year | Dataset title | Dataset ID and/or URL | Database, license, and accessibility information |
|---|---|---|---|---|
| Sumanprava Giri, Vasudha Aggarwal, Julien Pontis, Zhen Shen, Arindam Chakraborty, Abid Khan, Craig Mizzen, Kannanganattu V Prasanth, Slimane Ait-Si-Ali, Taekjip Ha, Supriya G Prasanth | 2015 | U2OS H3K9me3 ChIPseq | http://www.ncbi.nlm.nih. gov/geo/query/acc.cgi? acc=GSE68129 | Publicly available at NCBI Gene Expression Omnibus (GSE68129). |

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
