## [Decision Letter]

Thank you for sending your work entitled “ORCA/LRWD1 Organizes Heterochromatin by Assembling Histone H3 Lysine 9 Methyltransferases on Chromatin” for consideration at *eLife*. Your article has been overall favorably evaluated by James Manley (Senior editor) and three reviewers, one of whom is a member of our Board of Reviewing Editors. The Reviewing editor, Michael Botchan, and the other reviewers discussed their comments before we reached this decision, and the Reviewing editor has assembled the following comments to help you prepare a revised submission.

The manuscript has important new data that could in principle be published in *eLife*. However as you will see from the comments below, a considerable amount of additional experiments will be required and some of your major conclusions may be difficult to substantiate. But we would like to see if you can provide such work in a revised manuscript.

The association of the Lysine methytransferase complexes with ORCA and ORC is important and ORCA's role as a scaffold for this megacomplex is certainly something of potential wide importance. The authors' conclusion that ORCA brings the histone modification enzymes to heterochromatin and that these modifications in turn effect replication within heterochromatin is also a key result of potential impact.

The novel and key biological relevance sections of this paper hinges on the idea that ORCA-mediated heterochromatin formation is affecting DNA replication. The trouble the reviewers all have with this conclusion that needs to be addressed is that previously Prasanth and colleagues have shown some rather profound S phase phenotypes from ORCA depletion that they originally argued were due to a direct recruitment of ORC to DNA via ORCA. This S phase defect is documented again here in Figure 7. In this submission, Figures 5 and 7 are replication-timing phenotypes from asynchronous ORCA-depleted cells which presumably, given previous data, have too little ORC on DNA. Thus these experiments can't distinguish between heterochromatin-mediated origin firing effects and indirect effects of just having too little bound ORC, which could have its own effects on chromatin (i.e. via HP1).

The main attempt to address this issue of what ORCA does to ORC vs. ORCA's effect on histone methylation is Figure 8; the description of this very key experiment didn't get the attention it deserves in the writing and lacks some important details. The idea, not at all clearly described, is to remove endogenous ORCA with siRNA (which takes days) but keep cells healthy with the DD ORCA that they can then degrade rapidly after G1. Then they arrest cells after S phase starts (not G1/S as they claim since they're using aphidicolin) so presumably ORC loading is fine, and then they endeavor to see what happens to the next S phase progression if ORCA is degraded. There are a lot of controls they didn't show and must be done for further consideration:

1) Is ORC binding in fact fine? And if so some key data supporting that conclusion is required.

2) Does the degradable ORCA construct complement si-ORCA-depleted cells? If that all works the way they intended, then what do the authors feel about the phenotype in Figure 8? Is it likely to be biologically important? The effects seem much smaller than in Figure 7 that is the asynchronous (and therefore combination) phenotype. The reviewers think the Figure 8 differences are really small, and can't tell what one is supposed to be noticing that's different in the micrographs of Figure 8 and Figure 8. Could almost all the heterochromatin effects they document in other figures be co-dependent on ORC?

3) A related but different issue is the following: there's a critical control missing that should compare the overall progression through S phase of ORCA-depleted cells to controls. Given the strong S phase reduction from ORCA depletion (Figure 7 and in the authors' 2010 paper), the changes in replication of loci queried at a given time point may appear different just because there's overall fewer origins firing. The details of the replication timing analysis are also missing from the methods; how were the signals normalized, and what exactly are the BrdU-ChIP plots showing? Some of the key points about ORCA scaffolding come only from experiments in which the proteins of interest are overproduced. Figure 6 is one of those, and all of the SiMPull assays are done with transient transfections of fluorescent fusion constructs. Given that this is their first use of SiMPull colocalization to claim that three proteins are in a complex together, and that this is one of their major findings, a second method should be added as corroboration (e.g. FRET, ip-/re-IP, etc.).

Another set of important and major considerations of concern to all of the reviewers is the fact that most of the experiments are with over-expressed proteins and very little confidence is provided by the quality of the western blots shown in Figure 1. Some quantification of levels of the endogenous complex will be required as indicated below. Furthermore, the manuscript if it is to be published in *eLife* should use some corroborating evidence beyond the elegant SimPul fluorescence images to show that there's an endogenous complex scaffolded by ORCA.

1) Figure 1 and 1Ab are endogenous co-IPs but the labeling and quality seems rather poor. Perhaps better antibodies or purification of the reagents will cut out the background and enhance the signal. From such improved westerns some indication of complex levels as opposed to simply free protein can be made.

2) The manuscript cites a previous paper that describes the KMT “mega complex”. If a major point here is that ORCA is the scaffold, then the data in this manuscript should be able to show that by knocking down ORCA and repeating the G9A/Suv39H1 co-IP that's shown in 1Ab with anticipated results.

3) The fluorescent single molecule experiments do all have to use ectopic expression, but the reviewers think it would be better if the manuscript had at least one experiment with controlled expression levels so that the fluorescent molecules are in the range of the endogenous proteins. They could easily do that with stable cell lines.

Another major concern of the reviewers focused on the replication timing experiments. There was agreement that this is a point that might deserve a more specific analysis. A much more thorough demonstration for the points claimed, including a genome-wide analysis of the replication timing, is required. All the replication timing data is from cells that have a serious S phase defect (Figure 7), and reviewers agreed that even though the authors have done a considerable amount of work on this aspect, they were not convinced that the conclusions were substantiated by the data. Do cells depleted of ORC to the same extent as the ORCA depletion have the same sort of replication timing phenotype? This comes back to the overall question of what are the direct effects and what are indirect that were discussed above.

If the authors are convinced that the model in Figure 9 is true, then they would predict that simply knocking down ORC should have little effect on heterochromatin formation.

Finally some internal discrepancies from the figures arise and are key to what is actually happening to histone modification levels genome wide and in heterochromatin specifically:

1) Figure 4 shows that ORCA depletion does not affect the total level of H3K9me3 (as measured by Western blotting). However, from the Chip-seq data (Figure 5), the authors conclude that the H3K9me3 signal is significantly reduced. The reviewers are making the assumption that the modification of H3K9 takes place on chromatin and therefore, do not understand this apparent discrepancy. The authors also claimed that the H3K9me3 signal was significantly reduced at satellite repeats, but this is less than obvious in Figure 5.

2) Upon ORCA depletion, the authors found that the reduction of H3K9me3 peaks is localized in late replicating domains. However, we point out that most H3K9me3 sites are normally in late replicating domains and therefore the reduction must basically “mainly” affect late replication domains.

Minor comments:

1) The effect of transcriptional activation on the LacI array (Figure 2 and associated supplemental figures) is very interesting, but is not easily reconciled with the evidence that ORCA binds G91 and Suv38H1 independently and directly (Figure 1). It may be that the direct interactions are very weak since the authors needed to use western blotting to detect them. If so, then the interactions may in fact be chromatin dependent in cells rather than just chromatin-influenced; the use of EtBr in Figure 1 doesn't sway me on this since those are all co-IPs from overproduced proteins. The interactions with purified proteins show that they proteins can interact directly, but Figure 2 suggests that in cells any direct interaction is too weak to detect. The authors should make it clear what they mean in their interpretation of Figure 2.

2) Some parts of the figures are quite hard to read without major magnification. This is most common for graphs and some gels; should there be visible signal in the inputs in Figure 5? Also, what is “Hela S3 G1b 1” and “HeLa S3 S1 1”? I assume one is the control and one is the ORCA knockdown, but that's not said in the text or legends, and there is not enough information to interpret the replication timing graphs.

---

## [Author Response]

*The novel and key biological relevance sections of this paper hinges on the idea that ORCA-mediated heterochromatin formation is affecting DNA replication. The trouble the reviewers all have with this conclusion that needs to be addressed is that previously Prasanth and colleagues have shown some rather profound S phase phenotypes from ORCA depletion that they originally argued were due to a direct recruitment of ORC to DNA via ORCA. This S phase defect is documented again here in*
Figure 7*. In this submission,*
Figures 5 and 7
*are replication-timing phenotypes from asynchronous ORCA-depleted cells which presumably, given previous data, have too little ORC on DNA. Thus these experiments can't distinguish between heterochromatin-mediated origin firing effects and indirect effects of just having too little bound ORC, which could have its own effects on chromatin (i.e. via HP1)*.

*The main attempt to address this issue of what ORCA does to ORC vs. ORCA's effect on histone methylation is*
Figure 8*; the description of this very key experiment didn't get the attention it deserves in the writing and lacks some important details. The idea, not at all clearly described, is to remove endogenous ORCA with siRNA (which takes days) but keep cells healthy with the DD ORCA that they can then degrade rapidly after G1. Then they arrest cells after S phase starts (not G1/S as they claim since they're using aphidicolin) so presumably ORC loading is fine, and then they endeavor to see what happens to the next S phase progression if ORCA is degraded. There are a lot of controls they didn't show and must be done for further consideration*:

We apologize for not providing adequate details of the experiment. We have included the details of the experiments in the Materials and methods section (please see the paragraph entitled “Depletion of ORCA using proteotuner”, under the “Cell culture” subsection).

*1) Is ORC binding in fact fine? And if so some key data supporting that conclusion is required*.

To evaluate if ORC loading is indeed fine during G1 phase when only DD-ORCA is expressing, we have evaluated the Orc2 loading in cells that are depleted of ORCA in the presence or absence of DD- ORCA (please see experimental details below, also included in the Materials and methods section). We deplete endogenous ORCA in cells having DD-ORCA. In the presence of Shield, DD-ORCA is stabilized (Figure 8). We observe that in the absence of ORCA, Orc2 loading to chromatin is reduced, with a concomitant increase in the soluble pool of Orc2 (Figure 8). However, in cells expressing DD-ORCA, when the endogenous ORCA is depleted, DD-ORCA is able to rescue the chromatin-loading defect suggesting that DD- ORCA is functional.

Since the key role of ORC in loading MCM is restricted to G1 phase of the cell cycle, we have not evaluated how ORC loading is affected in ORCA- depleted S-phase cells. Further, the role of ORC binding to chromatin in postG1 cells remains to be understood.

Rescuing of ORC loading by using DD-T7-ORCA-siRNA NTV: U2OS cells were transfected with of DD-T7- ORCA-siRNA NTV construct along with siRNA against ORCA. Five hours later Shield1 (0.5µM) was added to the medium. 24 hours after the first round of knockdown, a second round of ORCA siRNA treatment was carried out in the presence of Shield1. Samples were collected 24h later for chromatin fractionation to examine ORC loading.

2) Does the degradable ORCA construct complement si-ORCA-depleted cells?

In addition, to demonstrating that DD-ORCA can rescue chromatin loading of Orc2 in the absence of endogenous ORCA, we have now conducted additional experiments to evaluate the functionality of DD-ORCA. Immunoprecipitation using T7-antibody in cells expressing DD-T7-ORCA showed efficient association of the DD-ORCA with ORC (Figure 8—figure supplement 1).

*If that all works the way they intended, then what do the authors feel about the phenotype in*
Figure 8*? Is it likely to be biologically important? The effects seem much smaller than in*
Figure 7
*that is the asynchronous (and therefore combination) phenotype*.

We agree with the reviewers that this phenotype of reduction in mid and late S patterns upon loss of ORCA is very likely a combination phenotype. We believe that the decrease though small, is statistically significant. This might be a reflection of the multiple, additive layers of regulation exerted by ORCA on the process of preRC assembly and heterochromatin organization. One possibility could be that the defects in preRC assembly as well as changes in heterochromatin organization cause changes in replication timing. Therefore, in cells where preRC is assembled appropriately, and ORCA is depleted in postG1 cells, the defects in heterochromatin organization cause defects in replication timing, but not to the same extent when initiation is also affected.

*The reviewers think the*
Figure 8
*differences are really small, and can't tell what one is supposed to be noticing that's different in the micrographs of*
Figure 8
*and*
Figure 8*. Could almost all the heterochromatin effects they document in other figures be co-dependent on ORC?*

We apologize that our description was not clear. We are now also providing higher magnification of these images (Figure 8 and Figure 8). We would like to bring to the reader’s attention that in the absence of ORCA, there is accumulation of BrdU and HP1a and H3K9me3 in the form of perinucleolar rings (Figure 8). Such a phenotype is reminiscent of the defects observed in HP1a and H3K9me3 distribution in Orc1 and Orc5 depleted human cells ([34], Supplementary figure).

We have demonstrated earlier, that ORC binding to chromatin is abrogated in the absence of ORCA ([44] Mol. Cell). Also, ORCA degrades in the absence Orc2 ([44] Cell Cycle). So, we know that ORCA cannot exist without Orc2. In the current manuscript also, we have shown that ORCA- ORC-HMTs are in one single complex and that all the heterochromatin effects are indeed co-dependent on ORC.

To understand, whether this role of ORCA/ORC in heterochromatin organization was independent of its role in preRC assembly, we carried out the experiments presented in Figure 8 of the manuscript and that makes us believe that ORCA/ORC have a distinct role in heterochromatin organization beyond a role in preRC assembly.

*3) A related but different issue is the following: there's a critical control missing that should compare the overall progression through S phase of ORCA-depleted cells to controls*.

We have now carried out flow cytometry in our proteo-tuner experimental set up, where ORCA is depleted in post-G1 cells. FACS profile in control and ORCA-depleted cells at 0, 4, 8 and 12 h did not show any changes in the length of S-phase and the progression through S-phase was comparable in control and in cells lacking ORCA (Figure 8—figure supplement 1).

*Given the strong S phase reduction from ORCA depletion (*Figure 7
*and in the authors' 2010 paper), the changes in replication of loci queried at a given time point may appear different just because there's overall fewer origins firing*.

We have now provided details of our BrdU ChIP experiment and also included detailed discussion and interpretation related to this part (please see details below). Our BrdU ChIP experiment shows that there are major transitions (late to early and early to middle) in replication timing. It is true that there are fewer origins firing because ORC loading is affected upon ORCA KD, when ORCA is depleted from asynchronously growing cells. But, if that were the only effect on replication, then for a given locus, the BrdU ChIP plot would have shown exactly the same trend for control and knockdown experiments. The difference would have been that the heights of the plots would have significantly reduced in ORCA KD samples. Instead, what we observe is that the pattern of the plot changes dramatically with loci now showing a replication timing profile strikingly different from control with large scale late to early (Figure 7—figure supplement 1) and early to mid (Figure 7—figure supplement 1) transitions.

The details of the replication timing analysis are also missing from the methods; how were the signals normalized, and what exactly are the BrdU-ChIP plots showing?

We plot the H3K9me3 and replication timing enrichment sites over the genome (black bars). We used our H3K9me3 data set in Control condition and the Hela replication timing data from Encode consortium. This information has now been included in the manuscript as described below.

BrdU ChIP after ORCA knockdown and data analysis: two rounds of ORCA knockdown were carried out, 24h apart. The cells were then arrested using aphidicolin for 12h followed by release into S phase and samples were collected 0, 4, 8 and 12h post release for BrdU ChIP. Prior to each time point collection, cells were pulsed for 2h with BrdU (10µM).

Cells for each time point were then lysed with 300ul of SDS lysis buffer (1% SDS, 10mM EDTA, 50mM Tris pH 8.0). The lysates were subjected to sonication with Bioruptor Power- up (Diagenode). 100ul sheared chromatin aliquots were then placed on 95°C heat block for 10 minutes. This was followed by snap chilling the samples for 10 minutes. The samples were then diluted, precleared and processed further in a manner identical to the ChIP protocol described in the previous section.

qPCRs were carried out with purified DNA of input, BrdU ChIP and mouse IgG ChIP samples obtained at 0, 4, 8 and 12h post aphidicolin release timepoints. The qPCR signals of BrdU and mouse IgG samples were calculated as percent input values. Fold enrichment of BrdU ChIP over mouse IgG ChIP was calculated for each time point.

*Some of the key points about ORCA scaffolding come only from experiments in which the proteins of interest are overproduced.*
Figure 6
*is one of those, and all of the SiMPull assays are done with transient transfections of fluorescent fusion constructs. Given that this is their first use of SiMPull colocalization to claim that three proteins are in a complex together, and that this is one of their major findings, a second method should be added as corroboration (e.g. FRET, ip-/re-IP, etc.)*.

Our collaborators who developed this technology, have just recently, also published the use of single-molecule pull down to establish multiple components within a complex (Jain et al., PNAS Dec 2014 111(50):17833-38). In their manuscript, as a proof of principle they demonstrated the existence of the triple complex using Flag-mTOR, mCherry-Raptor and YFP-PRAS40.

As per the reviewers’ suggestions, we have now performed IP/re-IP and provide evidence for the existence of the complex containing ORCA- ORC-G9a (Figure 3—figure supplement 1) and ORCA- G9a-Suv39h1 (Figure 3—figure supplement 1).

*Another set of important and major considerations of concern to all of the reviewers is the fact that most of the experiments are with over-expressed proteins and very little confidence is provided by the quality of the western blots shown in*
Figure 1*. Some quantification of levels of the endogenous complex will be required as indicated below. Furthermore, the manuscript if it is to be published in eLife should use some corroborating evidence beyond the elegant SimPul fluorescence images to show that there's an endogenous complex scaffolded by ORCA*.

*1)*
Figure 1
*are endogenous co-IPs but the labeling and quality seems rather poor. Perhaps better antibodies or purification of the reagents will cut out the background and enhance the signal. From such improved westerns some indication of complex levels as opposed to simply free protein can be made*.

We have evidence that ORCA IP and G9a IP can deplete ORCA and G9a respectively to completion (Figure 1—figure supplement 1). Based on 7 independent experiments, we have now estimated that 1.31% of total G9a is in a complex with ORCA. Similarly, 1.44% of total Suv39H1 is in a complex with ORCA (N=4). It has previously been reported that about 0.2% of endogenous H3K9 HKMTs co-purified with Suv39h1 (14). We are also providing two new sets of endogenous IP data to demonstrate that ORCA associates with G9a and that this interaction is not affected by the exogenous expression of FL-and SET (151-412)-Suv39H1 (Figure 1—figure supplement 1). Similarly, ORCA associates with Suv39h1, and this interaction is not affected by the exogenous expression of FL or Ankyrin-G9a (Figure 1—figure supplement 1).

*2) The manuscript cites a previous paper that describes the KMT “mega complex”. If a major point here is that ORCA is the scaffold, then the data in this manuscript should be able to show that by knocking down ORCA and repeating the G9A/Suv39H1 co-IP that's shown in 1Ab with anticipated results*.

We would like to bring it to the reviewers’ kind attention, that we need 30X10cm plates to conduct endogenous IPs (this is equivalent to 3X10e8 cells). As also pointed out in the original manuscript on the KMT megacomplex, ∼3 billion cells expressing FLAG-HA- Suv39H1 were required for a single glycerol gradient sedimentation to obtain the H3K9KMT signals detectable by western blotting (14). Conducting siRNA knockdown of 30X10cm plates for control and additional 30X10cm for ORCA knockdown (and attempt this thrice) is extremely difficult, expensive and an arduous task. We did however attempt this experiment without success. Because of this problem, we carried out SiMpull, since this can be achieved with 1 million cells. I hope the reviewers’ understand and agree with the logistics relating to this experiment.

*3) The fluorescent single molecule experiments do all have to use ectopic expression, but the reviewers think it would be better if the manuscript had at least one experiment with controlled expression levels so that the fluorescent molecules are in the range of the endogenous proteins. They could easily do that with stable cell lines*.

We thank the reviewers for this and we have now carefully titrated the levels of plasmid transfected and the levels of expression. We have now carried out SimPull using transiently transfected proteins whose levels are carefully titrated and are comparable to endogenous levels. Kindly note that we have now used only 100ng of each plasmid (T7-ORCA, mCherry- G9a and YFP-Suv39H1; Figure 3—figure supplement 1) and found results similar to what we reported previously (Figure 3). Similarly, we have now used 100ng of T7ORCA, mCherry-G9a and YFP-Orc1 and conducted SimPull analysis (Figure 3—figure supplement 1 and Figure 3). We have now replaced these figures in the manuscript.

In order to look at the formation of ORCA-G9a- Suv39H1 triple complex in a more endogenous context, we carried out SiMPull in a HA- ORCA U2OS stable cell line established in our lab. Towards this end, we transfected YFP- Suv39H1 and mCherry-G9a into the stable cell line and carried out SiMPull with both HA and ORCA antibody. Unfortunately, the signal to noise ratio was very low in both the pull-downs making further analysis of co-localization difficult (Figure 10). In addition, we also utilized a Flag-HA-Suv39H1 HeLa stable cell line into which we transfected YFP-ORC1 and mCherry-G9a. We then carried out SiMPull using Flag antibody. While mCherry-ORCA was pulled down efficiently by this process, YFP-ORC1 pulldown showed a very low signal to noise ratio, again making co-localization analysis difficult.

Author response image 1.**DOI:**
http://dx.doi.org/10.7554/eLife.06496.023

*Another major concern of the reviewers focused on the replication timing experiments. There was agreement that this is a point that might deserve a more specific analysis. A much more thorough demonstration for the points claimed, including a genome-wide analysis of the replication timing, is required. All the replication timing data is from cells that have a serious S phase defect (*Figure 7*), and reviewers agreed that even though the authors have done a considerable amount of work on this aspect, they were not convinced that the conclusions were substantiated by the data*.

*Do cells depleted of ORC to the same extent as the ORCA depletion have the same sort of replication timing phenotype? This comes back to the overall question of what are the direct effects and what are indirect that were discussed above*.

As pointed earlier, ORCA cannot exist in cells without ORC and therefore we believe that ORCA-ORC complex is critical for the heterochromatin organization function.

Kindly note that loss of Orc2 in human HeLa cells also shows changes in replication patterning. As documented previously, during a 10 min pulse in HeLa cells, 26% cells that incorporate BrdU, show 60% early, 25% mid and 15% late S pattern. In contrast, in Orc2 siRNA treated cells, 15% cells incorporate BrdU, out of which 75% represent early, 19% in mid and 6% are late S phase ([33] EMBO).

*If the authors are convinced that the model in*
Figure 9
*is true, then they would predict that simply knocking down ORC should have little effect on heterochromatin formation*.

Since ORCA-ORC is always in one complex, our model supports that this complex binds to KHMTs and together establish heterochromatin organization. We apologize that this was not clearly mentioned in the previous version. We have accordingly made the necessary changes to the manuscript.

*Finally some internal discrepancies from the figures arise and are key to what is actually happening to histone modification levels genome wide and in heterochromatin specifically*:

*1)*
Figure 4
*shows that ORCA depletion does not affect the total level of H3K9me3 (as measured by Western blotting). However, from the Chip-seq data (*Figure 5*), the authors conclude that the H3K9me3 signal is significantly reduced. The reviewers are making the assumption that the modification of H3K9 takes place on chromatin and therefore, do not understand this apparent discrepancy*.

We too have been baffled by this observation. Please note that we observe 18% reduction in H3K9me3 binding to chromatin upon ORCA knockdown. The explanation, we believe is that this reduction may not be detectable at the western blotting level.

*The authors also claimed that the H3K9me3 signal was significantly reduced at satellite repeats, but this is less than obvious in*
Figure 5.

As the reviewers pointed out, we observe approximately 15% decrease in H3K9me3 on total satellite repeats. This small but significant decrease is consistent with reports, which show marginal decrease of H3K9me3 marks at pericentromeric heterochromatin upon loss of critical regulatory factors (7). For example, upon loss of Pax3 or Pax9, there is a temporary decrease in H3K9me3 at heterochromatin from which the cell recovers in a couple of generations. As a result, only a marginal reduction of H3K9me3 is observed at mouse major satellites after extended knockdown of these transcription factors (7). While that is the case with knockdown of Pax3 and Pax9 individually, the results are very dramatic upon knocking down of Pax3 and Pax6 together, indicating cooperativity.

In addition, while the analysis H3K9me3 marks on total satellite repeats shows a small reduction, there is approximately a 50% reduction of this mark at specific telomeric (TAR1) and centromeric (SST1) repeats, indicating that the reduction of repressive marks could possibly be more pronounced at certain genomic loci as compared to others.

*2) Upon ORCA depletion, the authors found that the reduction of H3K9me3 peaks is localized in late replicating domains. However, we point out that most H3K9me3 sites are normally in late replicating domains and therefore the reduction must basically “mainly” affect late replication domains*.

We agree with the reviewer completely.

Minor comments:

*1) The effect of transcriptional activation on the LacI array (*Figure 2
*and associated supplemental figures) is very interesting, but is not easily reconciled with the evidence that ORCA binds G91 and Suv38H1 independently and directly (*Figure 1*). It may be that the direct interactions are very weak since the authors needed to use western blotting to detect them. If so, then the interactions may in fact be chromatin dependent in cells rather than just chromatin-influenced; the use of EtBr in*
Figure 1
*doesn't sway me on this since those are all co-IPs from overproduced proteins. The interactions with purified proteins show that they proteins can interact directly, but*
Figure 2
*suggests that in cells any direct interaction is too weak to detect. The authors should make it clear what they mean in their interpretation of*
Figure 2.

We agree with the reviewer and we have modified the text and discussed the relevant results in the above-mentioned manner.

*2) Some parts of the figures are quite hard to read without major magnification. This is most common for graphs and some gels; should there be visible signal in the inputs in*
Figure 5*? Also, what is “Hela S3 G1b 1” and “HeLa S3 S1 1”? I assume one is the control and one is the ORCA knockdown, but that's not said in the text or legends, and there is not enough information to interpret the replication timing graphs*.

We apologize. We have now increased the font size of the labeling. We are also providing a better intensity image for 5A (Figure 5).

Initial analysis of the available repli-seq dataset from various human cell lines in UCSC Genome Browser and ENCODE consortium revealed that the replication timing of large domains remain the same across cell lines. We therefore compared the H3K9me3 ChIP-seq data set to the HeLa repli-seq dataset. HeLa-S3 G1b and HeLa-S3 S1 are deep sequencing data sets for late G1 and early S replicating regions in HeLa-S3 cells (16). The chromosomal regions that are replicating at these two stages are shown in black (early) and gray (late) along the length of the chromosome.

Using the dataset mentioned above, we examined the replication timing of the regions that showed reduction in H3K9me3 by ChIP-seq. On chromosome 19, the total H3K9me3 peaks in the control sample and the regions, which show greater than 5 fold decrease in H3K9me3 upon ORCA depletion are represented as black bars above the HeLa-S3 G1b and HeLa-S3 S1 tracks. Upon ORCA depletion, most of the affected H3K9me3 peaks resided in late replicating domains. This coupled with the loss of late replication patterns by BrdU IF in ORCA-depleted cells made us hypothesize that ORCA could also regulate the replication of late replicating regions.

We have included the above details in the text of the revised manuscript.